# Verification of parameterizations for clear sky downwelling longwave irradiance in the Arctic

Giandomenico Pace[1], Alcide di Sarra[2], Filippo Cali Quaglia[3,4], Virginia Ciardini[1], Tatiana Di Iorio[1], Antonio Iaccarino[1], Daniela Meloni[1], Giovanni Muscari[3], Claudio Scarchilli[1]

[1]Laboratory for Observations and Measurements for Environment and Climate, ENEA, Rome, 00123, Italy
[2]Laboratory for Observations and Measurements for Environment and Climate, ENEA, Frascati, 00044, Italy
[3]INGV, Rome, 00143, Italy
[4]Ca' Foscari University of Venice, Mestre-Venezia, 30172, Italy

*Correspondence to*: Giandomenico Pace (giandomenico.pace@enea.it)

**Abstract.** Ground-based high resolution observations of downward longwave irradiance (DLI), surface air temperature, water vapour surface partial pressure and column amount, zenith sky infrared (IR) radiance in the atmospheric window, and all-sky camera images are regularly obtained at the Thule High Arctic Atmospheric Observatory (THAAO, 76.5°N, 68.8°W), North-West Greenland. The datasets for the years 2017 and 2018 have been used to assess the performance of different empirical formulas used to infer clear sky DLI. An algorithm to identify clear sky observations has been developed, based on value, variability, and persistence of zenith sky IR radiance. Seventeen different formulas to estimate DLI have been tested against the THAAO dataset, using the originally determined coefficients. The formulas that combine information on total column water vapour and surface air temperature appear to perform better than others, with a mean bias with respect to the measured DLI smaller than 1 W/m² and a root mean squared error (RMSE) around 6 W/m². Unexpectedly, some formulas specifically developed for the Arctic are found to produce poor statistical results; this is attributed partly to limitations in the originally used dataset, which does not cover a whole year or is relative to very specific conditions (i.e., the presence of a ice sheet ). As expected, the bias displays a significant improvement when the coefficients of the different formulas are calculated using the THAAO dataset. The presence of two full years of data allows the determination and the applicability of the coefficients for singular years and the evaluation of results. The smallest values of the bias and RMSE reach 0.1 W/m² and 5 W/m², respectively. Overall, best results are found for formulas that use both surface parameters and total water vapour column content, and have been developed from global datasets. Conversely, formulas that express the atmospheric emissivity as a linear function of the logarithm of the column integrated water vapour appear to reproduce poorly the observations at THAAO.

## 1 Introduction

The Arctic region is showing the most intense warming of the globe, because of different regional feedback mechanisms often related to the sea ice decline(Taylor et al. 2022) . Both observed and projected warming rates reach a maximum in the autumn and winter seasons (Bintanja and Krikken, 2016), when the Arctic surface energy budget is dominated by longwave radiation.

Indeed, due to the large seasonal variation of shortwave radiation, the longwave radiation plays a key role in the Arctic, where the annual total Downward Longwave Irradiance (DLI) is usually more than twice as large as the annual downward shortwave irradiance (Curry et al., 1996). Nevertheless, surface longwave irradiance measurements in the Arctic are particularly scarce, and retrievals of surface radiation budget based on satellite data are notoriously problematic at high latitudes (e.g., Kay and L'Ecuyer, 2013; Di Biagio et al., 2021).

In clear sky conditions, DLI is determined by the atmospheric concentration of the main greenhouse gases (i.e., principally water vapor but also carbon dioxide, methane, and ozone) and by their radiating temperature. Additionally, Gupta (1989) and Ohmura (2001) showed that about 86% of surface DLI is originating between 1000 and 900 mb, and 95% between 1000 and 700 mb, indicating that the main contribution to the surface DLI is related to the composition and temperature of the lowest atmospheric layers. Downward longwave irradiance is also strongly affected by the presence of clouds and by their characteristics. Clouds generally induce an increase of surface DLI which is modulated by cloud cover, altitude, phase, water or/and ice concentration, and cloud particle number and size (Shupe and Intrieri 2004).

Methods to estimate DLI include the use of complex radiative transfer models or semi-empirical formulas based on available local measurements of atmospheric parameters, the former requiring detailed knowledge of the atmospheric vertical structure and composition. However, these are often not available, especially in the Arctic region (Key et al., 1996) where regular radiosoundings are scarce. Due to these limitations, various semi-empirical formulas have been developed, and some specifically for the Arctic.

Most of the semi-empirical formulas aim at the DLI estimation in clear sky conditions. This is particularly important because the clear sky DLI is needed to quantify the longwave radiative effect of clouds or other atmospheric components on the surface DLI. Starting from the pioneering work of Ångström (1918), several formulas for clear sky DLI have been developed, mostly to parameterize the clear sky atmospheric effective emissivity ($\varepsilon$). These formulas use surface or columnar atmospheric measurements, such as screen-level air temperature (i.e., air temperature at 2 m above ground), or/and water vapour partial pressure and/or integrated water vapour. The parameterizations in Dürr and Philipona (2004) and Long and Turner (2008) differ from those considered in this work because they use explicit dependences on the annual and daily variability of the observed atmospheric parameters and DLI at the measurement site and therefore require specific analyses. Both the works improve the parameterization of $\varepsilon$ presented by Brutsaert (1975) by refining the estimation of the so-called Lapse Rate Coefficient. Dürr and Philipona (2004) approximated the diurnal and annual cycle of the considered sites using a periodical function, while Long and Turner (2008) analyzed separately the daytime and nighttime behavior of the Lapse Rate Coefficient interpolating the daily results during sunset and sunrise; they also applied this method to the Arctic site of North Slope in Alaska, finding differences within ± 4 W/m$^2$ between the measured and observed DLI values in 68% of cases.

Few authors have carried out extensive comparisons among the different formulas for clear sky DLI. Flerchinger et al. (2009) tested 13 different formulas to estimate the DLI under clear sky conditions and 4 formulas for all-sky conditions using data from 21 sites across North America and China. Formetta et al. (2016) evaluated the performance of 10 different formulas using

both literature and site optimized coefficients taking into account data from 24 stations across the USA, chosen among the 65 stations of the AmeriFlux Network. More recently, Yang et al. (2023) used a long-term hourly database of DLI and meteorological parameters acquired between 2011 and 2022 at 7 stations of the China Baseline Surface Radiation Network to evaluate the performances of 3 different DLI formulas, both in clear and all-sky conditions. These authors, as well as others, showed that a site-specific calibration of the formulas used for the DLI estimates strongly improves their performances.

Some studies (e.g., Hanesiak et al., 2001; Niemelä et al., 2001; Jin et al., 2006) tested clear sky formulations specifically for Artic conditions confirming the need to optimize the DLI formulas also in this region.

The main objective of this paper is to investigate the performance of published and site-optimized DLI formulas in the Arctic environment by means of continuous measurements of surface DLI, screen-level temperature, and water vapor partial pressure, as well as integrated water vapor, obtained at the Thule High Arctic Atmospheric Observatory (THAAO; 76.5°N, 68.8°W; http://www.thuleatmos-it.it/), in North-Western Greenland. The analysis uses two full years (2017 and 2018) of observations carried out at THAAO.

The paper is organized as follows. Section 2 describes the site and the measurements used in the analysis, while section 3 discusses the methodology used to identify the clear sky periods selected for the analysis. The atmospheric conditions occurring during the two years and the statistical indices used to evaluate the performances of the DLI formulas are discussed in section 4. Section 5 briefly reviews the DLI formulas used in this study and in Section 6 the results of both original (published) and Pituffik optimized formulas are discussed. Conclusions are reported in Section 7.

## 2 Site and measurements

This study uses measurements of surface meteorological parameters, DLI, infrared zenith sky brightness temperature (IBT) in the 9.6-11.5 μm spectral range, and integrated water vapour (IWV) carried out at THAAO during 2017 and 2018. The THAAO is located on South Mountain, at 220 m a.s.l., near the Pituffik Space Base (formerly known as Thule Air Base), along the north-western coast of Greenland at about 3 km from the sea and 11 km from the Greenland ice sheet (GrIS). Therefore, the THAAO environment is typical of the northern coastal area of Greenland, i.e., influenced by both the GrIS which generates strong katabatic winds, and by the sea, especially in summer when open waters prevail over sea ice. Pituffik is also located in a region, which includes the area northwest of Greenland and the Ellesmere Island, characterized by an atmosphere particularly dry (Cox et al. 2012), with higher, colder and thinner clouds with respect to what is found in other areas of the Arctic (Shupe et al. 2011).

The measurements at THAAO are part of a long-term effort dedicated to the investigation of the Arctic climate. Studies on the evolution of the Arctic polar vortex (e.g., di Sarra et al., 2002; Muscari et al., 2007; Di Biagio et al., 2010; Mevi et al., 2018), aerosol/water vapor/albedo feedbacks (Di Biagio et al., 2012), aerosol properties (Becagli et al, 2016, 2019, 2020; Calì Quaglia et al., 2022), surface radiation (Muscari et al., 2014; Calì Quaglia et al., 2022,Meloni et al. 2023) were carried out based on measurements at THAAO. Additional instruments, whose observations are used in this analysis, were installed in July 2016

and have been operational since then. The roof of the THAAO building has a clear horizon, free of obstacles, and is a good site for high quality radiation measurements.

Pressure, screen-level temperature and relative humidity (P, $T_s$ and $RH_s$) are measured by means of a Campbell weather station (temperature and humidity through a HC2-S3 probe) installed on the roof of the THAAO building, ~4 m above the ground, and are collected every 10 minutes until the end of January 2022, every minute afterwards by means of a datalogger CR200X. The water vapor pressure at the surface level ($e_s$, in hPa) has been calculated using the values of relative humidity and of the saturation pressure ($e_{sat}$, in hPa) formulas with respect to water and ice following Wagner and Pruß (2002).

DLI and IBT are measured by a ventilated Kipp&Zonen CGR4 pyrgeometer and by a Heitronics KT19.85 II Infrared Radiation Pyrometer, respectively, with their signals recorded every minute using a datalogger.

The pyrgeometer was calibrated by the manufacturer in 2012, and its calibration has been verified by comparison with freshly calibrated pyrgeometers, traceable to the World Infrared Standard Group (WISG; WMO 2014), in 2013, 2016, and 2019. The CGR4 sensitivity was determined down to -40°C in order to obtain reliable measurements also at very low temperatures. The expanded uncertainty on DLI measurements is estimated to be ±5 W m$^{-2}$ (Meloni et al., 2012).

The KT19.85 pyrometer was modified by the manufacturer to extend the measurement range from -50/200°C to -150/300 °C. The instrument is calibrated down to -100 °C and has a response linearity that deteriorates between -100 and -150 °C. The IRT is installed on the roof looking at the zenith; air from inside the building is continuously blown on the pyrometer external window to prevent the formation/deposition of ice and snow. Visual inspections and quality data control confirm that this solution is very effective in keeping the pyrometer external window clean. The pyrometer has a FOV of about 2.6° and its accuracy is estimated to be ±0.5 K + 0.7% of the temperature difference between the instrument body and the observed target (IRP Operational Instructions, 2008). For the operational conditions at THAAO the overall accuracy in the calibrated range is therefore ±1.5 K.

Integrated water vapour is retrieved from brightness temperature values measured at seven different frequencies in the K band, between 22 and 31 GHz by an RPG HATPRO-G2 microwave radiometer (i.e. MWR, Rose et al., 2005). IWV is derived from zenith brightness temperature measurements sampled every 2 seconds and averaged over a minute. The expected accuracy (indicated as root mean squared error, RMSE) on IWV is indicated by the manufacturer to be ± 0.2 mm (or kg/m$^2$). Thirty-five Vaisala RS92 radiosondes were launched from THAAO in the period July 2016 – February 2017 (23 in summer and 12 in winter). The mean bias and standard deviation between the values of IWV calculated from the radiosoundings and those retrieved by the MWR are -0.18 mm and 0.35 mm, respectively, confirming the good performance of the developed MWR retrieval (Pace et al., 2017).

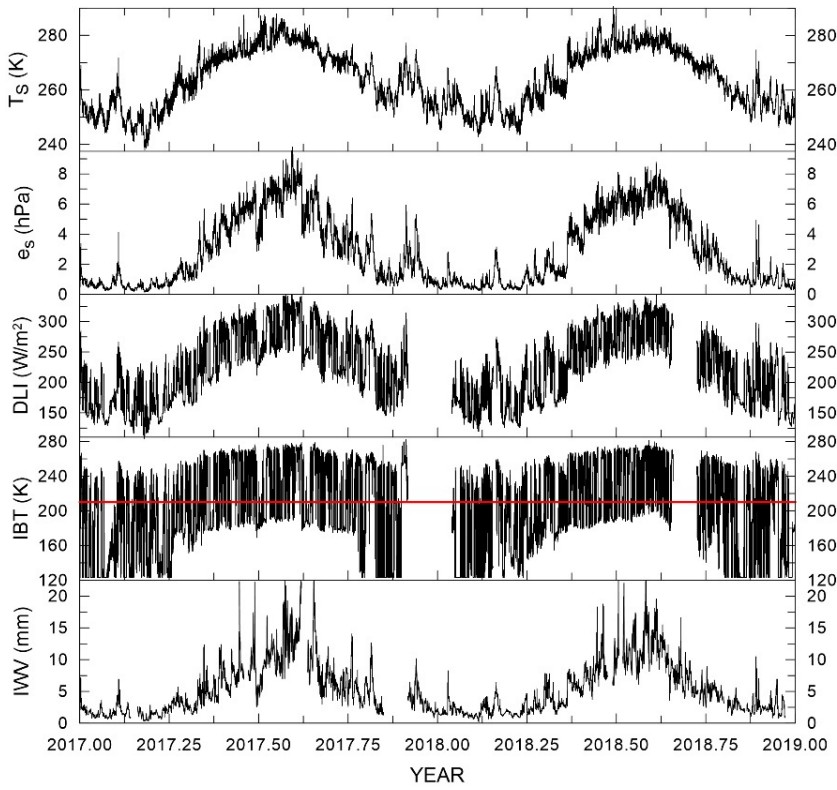

**Figure 1. From top to bottom: time series of $T_s$, $e_s$, DLI, IBT, and IWV for the years 2017 and 2018. Two major data gaps for DLI and IBT at the end of 2017 and in summer 2019 are due to instrumental problems. The red horizontal line in the IBT plot indicates the level of 210 K, that is considered the highest brightness temperature attainable in clear sky conditions (see text).**

The surface meteorological measurements have been linearly interpolated at the 1 minute acquisition frequency of the pyrgeometer and pyrometer data.

Time series of $T_s$, $e_s$, DLI, IBT, IWV are shown in Figure 1. The annual cycle of all parameters is clearly visible. The periods with large high frequency variability of DLI and IBT generally indicate the presence of clouds. Several spikes of IWV may be induced by episodes of rain and deposition of water over the MWR radome, that is occasionally not efficiently cleaned by the instrument blower. As will be discussed below, these data have been discarded as part of the clear sky selection procedure.

## 3 Clear sky screening procedure

An important part of this work is the methodology developed to select the clear sky periods to be used as reference to derive and test the formulas available and the newly conceived ones.

The pyrometer data, acquired simultaneously with data from the pyrgeometer, are used for the clear sky detection. The pyrometer signal is chosen because its spectral band is totally contained in the atmospheric window, and its signal is weakly dependent on IWV, but strongly dependent on clouds occurrence. Moreover, due to its narrow field of view, the pyrometer's signal is strongly dependent on the inhomogeneities usually associated with clouds. These characteristics are important for determining the presence of clouds and allow to determine clear sky conditions throughout the year, during both daytime and nighttime. One limitation of this technique is that the pyrometer, due to the specific geometry, is sensitive only to clouds falling into the instrument field of view, i.e. at the zenith. The adopted algorithm has been developed to circumvent also this limitation, as will be discussed below.

Optically thick clouds produce a significant increase of IBT, that is expected to be clearly discernible; conversely, the IBT enhancement is smaller for optically thin clouds. All clouds are expected to induce a significant increase of the high frequency signal variability, except for very homogeneous thick clouds. Thus, by selecting appropriate thresholds for IBT, its standard deviation, and posing some requirements on the persistence of the observed IBT with time, it is in principle possible to identify cloud-free conditions.The procedure developed to determine the clear sky periods is summarized in the flow-chart of figure 2 and is described below. The analysis is based on 1-minute IBT data. As first step, 15 minute averages of IBT, $(\overline{IBT})$, and the corresponding standard deviation ($\sigma_{IBT}$), have been calculated. The time interval of 15 minutes was chosen following Kassianov et al. (2005), who showed that the typical sky decorrelation time for hemispheric instruments like the pyrgeometer is of the order of 15 min.

Large values of $\sigma_{IBT}$ are associated with the presence of clouds. However, stratiform uniform clouds may produce low values of $\sigma_{IBT}$, and therefore the screening must take into account also $\overline{IBT}$.

All $\overline{IBT}$ values > 210 K have been considered influenced by clouds, throughout the year (see the red line in the IBT plot of Figure 1). An empirical expression for the threshold values on $\sigma_{IBT}$, $\sigma_{thr}$, which varies as a function of IBT, has been derived by looking at pyrometer data and simultaneous sky imager pictures, also taking into account that the signal-to-noise ratio of the pyrometer increases for decreasing IBT. The following expression for the threshold on the standard deviation has been used for IBT between 122.5 and 210 K:

$$\sigma_{thr} = 1 + \frac{10}{[IBT - 122.5]^{0.8}} \tag{1}$$

Although the pyrometer sensor is calibrated only above 173 K (-100 °C) and below this brightness temperature it is outside its linearity regime, below 173 K the sensor still responds to changes of zenith sky radiance, and the selection procedure based on $\sigma_{thr}$ turns out to be applicable.

Temporal intervals characterized by $\overline{IBT}$ and $\sigma_{thr}$ lower than the selected thresholds are called zenith clear sky cases (ZCSC), and are thought to be characterized by clear sky conditions at the zenith over the pyrometer.

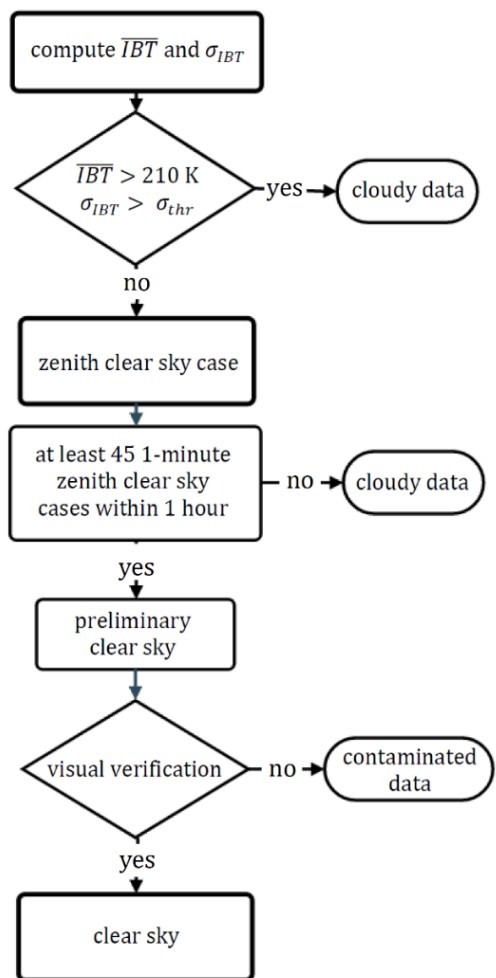

**Figure 2. Schematic view of the methodology adopted to identify clear sky conditions (see text).**

175

The identification of clear sky cases, however, requires that there are no clouds in the sky, and not only at the zenith. Broader time intervals, therefore using variability in time as a proxy for spatial variability, have been considered to infer clear sky conditions. Each IBT measurement at 1-minute resolution identified as ZCSC has been compared with those obtained in the previous and following 30 minutes; the ZCSC measurement has been considered a clear sky period if more than 45 individual

1-minute observations carried out during the 60 minutes interval were classified as ZCSC cases. This second condition is intended to identify clear sky conditions for hemispheric instruments such as the pyrgeometer. The choice to use the one-hour interval for the definition of clear sky is based on a preliminary analysis of the database, and it is in line with the approach followed by Dupont et al. (2008) who used hourly lidar averages, for comparing clear sky values derived from shortwave and longwave measurements with those derived from lidar measurements.

To evaluate the sensitivity of the implemented methodology to identify clear sky cases in the presence of thin cirrus clouds, both the IBT and the DLI were simulated for cloud-free conditions and with an homogeneous cirrus cloud, by means of the MODTRAN5.3 radiative transfer model (Berk et al., 2006). Different cloud optical thickness values were assumed, from 0.03 to 5, both in winter and summer conditions. The results of simulations are presented and discussed in the supplementary material. In general, the simulations highlighted the greater sensitivity of the pyrometer measurements compared to those of the pyrgeometer, particularly for low values of IWV. Based on our simulations a cirrus cloud with optical thickness of 0.1 covering homogeneously the sky in winter, induces an increase in the IBT e DLI signals compared to those for clear sky condition of 11.3 K and 2.7 W/m$^2$ respectively, corresponding to a percentage increase of 7.1% for the IBT and 1.6% for the DLI. For a similar summer case there would be an increase of 5 K and 2.7 W/m$^2$, respectively, corresponding to a percentage increase of 2.6% for the IBT and 0.97% for the DLI. These results confirm that the applied methodology is accurate enough to evaluate cases of ZCSC, even for cirrus clouds with optical thickness lower than 0.08-0.1. It should be noted that cirrus clouds of this optical thickness covering uniformly the sky induce variations in the DLI that are lower than the uncertainty of the DLI measurements, i.e. ±5 W/m$^2$, confirming that our clear sky methodology is sufficiently accurate to identify the DLI variation induced by clouds. The results of our simulations agree with those presented by Dupont et al. (2008), who highlighted that the DLI clear sky detection algorithm derived from DLI measurements perform correctly for cloud optical thickness of 0.3 or less, also evidencing that tall, thin clouds may not be detected by pyrgeometer measurements.A visual inspection of the results with respect to the sky imager pictures shows that this methodology is accurate, although it may fail in case of formation of snow/ice over the window or dome of the instruments.

The presence of snow/ice induces values of IBT larger than those expected for clear sky, but often < 210 K. At the same time, the snow/ice layer may produce a limited time variability of IBT, with $\sigma_{IBT}$ often below $\sigma_{thr}$, and a persistence of IBT values, thus satisfying the clear sky selection criteria. In order to remove these cases, all data identified as clear sky have been further subjected to a visual inspection, taking advantage of the sky imager pictures. The all-sky camera is not ventilated and is more subjected to the accumulation of snow/ice than the pyrometer and the pyrgeometer, that are both ventilated. Contaminated and dubious data, which are however a small fraction of the dataset, are discarded after visual inspection.

## 4 Data and metrics adopted to quantify the goodness of the different formulas

### 4.1 Dataset characteristics

As discussed above, data from 2017 and 2018 have been used in the analysis.

Figure 3 shows the distribution of the observed values of T$_s$, e$_s$, and IWV, during the two years. T$_s$ values show a bimodal distribution that is related to winter and summer seasons. In particular, 2018 measurements present a lower occurrence of intermediate values, resulting in a more pronounced separation of the two seasonal modes. The median value of T$_s$ is lower in 2018 with respect to 2017 (respectively 256.2 and 261.0 K in 2018 and 2017), although a larger occurrence of very low temperatures was observed in 2017. Consistently, values of e$_s$ < 0.3 hPa are more frequent in 2017 than in 2018.

Differences between the two years are present also in the occurrences of intermediate values of $T_s$ and $e_s$. The distribution of $e_s$ values in the two years appears in good agreement with that of IWV retrieved from MWR observations using independent information. Data from 2017 show lower extremes of IWV and an intermediate class around 3-5 mm, whereas 2018 measurements show larger occurrences of IWV in the range of 2-3 mm and between 6 and 7.5 mm. Winter 2017 thus was relatively colder and drier than winter 2018 and displayed larger maximum and minimum values.

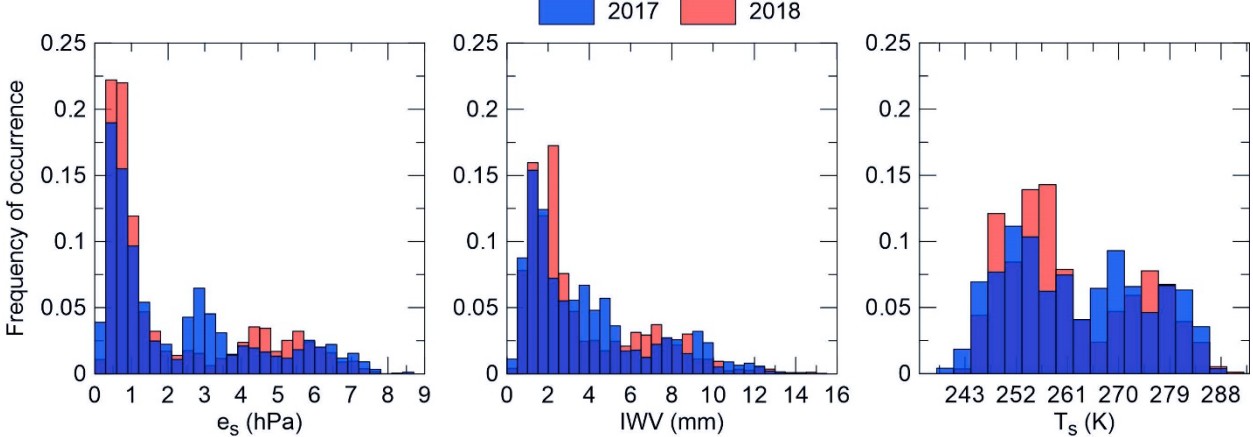

Figure 3. Annual frequency of occurrence of the measured values of $e_s$, IWV, and $T_s$ at THAAO in 2017 and 2018.

The inter-annual differences make the dataset suitable to investigate the applicability of the formulas to somewhat different conditions. As will be discussed below, we will use data from one year to derive coefficients of the different formulas specific for THAAO, and data from the second year to verify the results.

**4.2 Statistical indices**

Different metrics have been used in the literature to assess the performance of the formulas developed to estimate $\varepsilon$ or DLI. Also the characteristics of the used datasets may differ, depending on the region, season, data averaging interval, and on the source (direct measurements or radiative transfer model outputs). In most cases, measurements are used as reference for the determination of the coefficients appearing in the different formulas, but in some cases radiative transfer calculations have been used instead (e.g., Dilley and O'Brien, 1998).

With the aim of providing comparable results between our work and previous studies, a wide set of statistical indices were calculated for the different formulas. The statistical indices were calculated starting from 1 minute averages of clear sky DLI measurements ($m_i$) and DLI estimates ($p_{i,j}$), predicted by the different formulas (indicated by the j index) and calculated by using values of $T_s$, $e_s$, and IWV concurrent with $m_i$. For each formula, $d_{i,j} = p_{i,j} - m_i$ is defined as the difference between predicted and measured values of clear sky DLI.

The derived indices are the bias, the standard deviation, the root mean square error (RMSE), the skewness, the kurtosis and the $5^{th}$, $25^{th}$, $50^{th}$, $75^{th}$, and $95^{th}$ percentile of the difference $d_{i,j}$, as well as the squared linear Pearson correlation coefficient ($R^2$), and the slope of the of $p_{i,j}$ - $m_i$ linear fit.

Following Staiger and Matzarakis (2010) and Formetta et al. (2016) also the skill (hereafter named $T_{skill}$, Taylor, 2001) and the Kling-Gupta Efficiency (KGE, Gupta, 2009) have been determined to estimate the performance of the examined algorithms. The considered indices are defined as follows:

$$bias_j = \frac{1}{n} \sum d_{i,j} \tag{2}$$

$$STD_j = \sqrt{\frac{1}{n-1} \sum_i (d_{i,j} - \bar{d}_j)^2} \tag{3}$$

$$RMSE_j = \sqrt{bias_j^2 + STD_j^2} \tag{4}$$

$$Skewness_j = \frac{1}{n} \sum_i \left(\frac{d_{i,j} - \bar{d}_j}{STD_j}\right)^3 \tag{5}$$

$$Kurtosis_j = \frac{1}{n} \sum_i \left(\frac{d_{i,j} - \bar{d}_j}{STD_j}\right)^4 - 3 \tag{6}$$

The values of the kurtosis is an indication of the tail of the distribution: a normal distribution produces a kurtosis equal to 3, higher values are ascribed to a smaller number of large outliers with respect to the normal distribution. KGE includes the effects of correlation, bias, and variability and is expressed as

$$KGE_j = 1 - \sqrt{(r_j - 1)^2 + (a_j - 1)^2 + (b_j - 1)^2} \tag{7}$$

where $r_j$ is the linear correlation coefficient for formula j, $a_j$ is the ratio between the standard deviations of $p_{i,j}$ and $m_i$, and $b_j$ is the ratio between the mean values of $p_{i,j}$ and $m_i$ .

The $T_{skill}$ index summarize the capability of the different formulas to reproduce the observations and is given by

$$T_{Skill\,j} = \frac{4 \cdot (1+r_j)^4}{(s_j + 1/s_j)^2 \cdot (1+r_0)^4} \tag{8}$$

where $r_j$ is the linear correlation coefficient obtained using formula j, $r_0$ is the maximum correlation assumed equal to 1, and $s_j$ is the ratio between the variances of $p_{i,j}$ and $m_i$.

About 100,000 1 minute clear sky DLI measurements were selected and used in the analysis for each of the two years.

## 5 DLI parameterizations

Many algorithms estimating clear sky atmospheric effective emissivity or DLI from atmospheric meteorological parameters have been proposed, although only few of them have been tested in the Arctic environment. DLI for clear sky conditions is often expressed as

$$DLI = \varepsilon \, \sigma \, T_s^4 \tag{9}$$

where ε is the clear sky effective emissivity and σ is the Stefan-Boltzmann constant.  The clear sky effective emissivity depends on IWV and on other greenhouse gases, and is comprised between a value below 1 (corresponding to a dry atmosphere, with the lower limit determined by the concentration of other greenhouse gases) and 1 for a saturated atmosphere (e.g.,  Prata, 1996, and references therein). The formulas most commonly used in the literature and the few ones developed and tested for the

Artic region have been selected in this study. The used formulas for ε and DLI are summarized in Table 1.

**Table 1. List of the formulas for the clear sky effective atmospheric emissivity (ε) and the downward longwave irradiance (DLI) considered in this study, with an ID number identifying each formula. The source region of the data used to derive the coefficients is also indicated.**

| ID# | Formula | Geographical region | Reference |
|---|---|---|---|
| 1 | $\varepsilon = 0.7855$ | Barrow, Alaska, USA | Maykut and Church, 1973 |
| 2 | $\varepsilon = 0.67 + 0.05 \cdot e_s^{0.05}$ | Tiksi Bay, Jacutzia | Marshunova, 1966 |
| 3 | $\varepsilon = 9.365 \cdot 10^{-6} \cdot T_s^2$ | Northern Australia, Indian Ocean | Swinbank, 1963 |
| 4 | $\varepsilon = 1 - 0.261 \cdot \exp[-7.77 \cdot 10^{-4} (273 - T_s)^2]$ | Various climatic zones | Idso and Jackson, 1969 |
| 5 | $\varepsilon = 8.733 \cdot 10^{-3} \cdot T_s^{0.788}$ | Arctic | Ohmura, 1981 |
| 6 | $\varepsilon = 1.24 \cdot (e_s/T_s)^{1/7}$ | Mid-latitudes | Brutsaert, 1975 |
| 7 | $\varepsilon = 1.08 \cdot [1 - \exp(-e_s^{T_s/2016})]$ | Montana, Alaska, USA | Satterlund, 1979 |
| 8 | $\varepsilon = 0.70 + 5.95 \cdot 10^{-5} \cdot e_s \cdot \exp(1500/T_s)$ | Arizona, USA | Idso, 1981 |
| 9 | $\varepsilon = 0.0601 + 5.95 \, 10^{-5} \cdot e_s \cdot \exp(1500/T_s)$ | Arctic/Antarctica | Andreas and Ackley, 1994 |
| 10 | $\varepsilon = 0.23 + 0.484 \cdot (e_s/T_s)^{1/8}$ | Greenland ice sheet | Konzelmann et al., 1994 |
| 11 | $\varepsilon = [1.2983 - 0.0079 \cdot (T_s - 273.16) + 0.0003 \cdot (T_s - 273.16)^2] \cdot \left(\frac{e_s}{T_s}\right)^{1/7}$ | Arctic | Jin et al., 2006 |
| 12 | $\varepsilon = 1 - (1 + IWV) \cdot exp[-(1.2 + 3.0 \cdot IWV)^{0.5}]$ | Global data and radiation transfer model simulations | Prata, 1996 |
| 13 Zhang_A | DLI = $113.7 + 190.1 \cdot \ln(IWV)$ | Barrow, Alaska, USA | Zhang et al., 2001 |
| 14 Zhang_B | DLI = $125.6 + 104.6 \cdot \ln(IWV)$ | McGrath, Alaska, USA | Zhang et al., 2001 |
| 15 | $DLI = 155.12 + 48.75 \cdot \ln(IWV)$ | Canadian Arctic | Raddatz et al., 2013 |
| 16 Dilley_A | $\varepsilon = [1 - exp(-1.66\,\tau)]$ with $\tau = 2.232 - 1.875 \cdot \left(\frac{T_s}{273.16}\right) + 0.7356 \cdot \left(\frac{IWV}{IWV_0}\right)^{0.5}$ , with $IWV_0 = 25$ kg/m$^2$ | Global data and radiation transfer model simulations | Dilley and O'Brien, 1998 |
| 17 Dilley_B | $DLI = 59.38 + 113.7 \cdot \left(\frac{T_s}{273.16}\right)^6 + 96.96 \cdot \left(\frac{IWV}{IWV_0}\right)^{0.5}$ , with $IWV_0 = 25$ kg/m$^2$ | Global data and radiation transfer model simulations | Dilley and O'Brien, 1998 |


The simplest DLI parameterization is based on the assumption of a constant value for ε. Maykut and Church (1973) derived a constant value for ε of 0.7855 from 5 years of observations at Barrow (Alaska); this value differs only by 2.7% from the one later proposed by König-Langlo and Augstein (1994) who derived a value of 0.765 using data from Artic and Antarctic stations. The formulation by Maykut and Church (1973) has been tested (ID# 1).

In some cases ε has been related with the surface water vapour partial pressure, following the parameterization proposed by Brunt (1932). Marshunova (1966) optimized the coefficients of the formula by Brunt based on monthly mean observations from different Arctic sites, and we used her expression in the analysis (ID# 2).

ID# 3, 4, and 5 are instead based on formulas that use only $T_s$ to estimate ε. To justify this approach, Deacon (1970) suggested that, due to the strong coupling between $T_s$ and IWV, an explicit dependence on humidity may not be necessary.

Various formulas (ID# 6-11) use different combinations of $e_s$ and $T_s$ to estimate ε. The coefficients used in the formulas by Satterlund (1979), Andreas and Ackley (1994), Konzelmann et al. (1994), and Jin et al. (2006), corresponding to ID # 7, 9, 10, and 11, respectively, were derived based on Arctic data. Although the formula of Jin et al. (2006) (ID# 11) depends explicitly only on $e_s$ and $T_s$, it takes also into account the temperature and water vapor lapse rates by means of an empirical relationship developed for Arctic sites.

More recently, various authors have used IWV to derive ε or DLI (e.g., ID#12. Prata, 1996; ID# 13 and 14, Zhang et al., 2001; ID#15, Raddatz et al., 2013), while Dilley and O'Brien (1998) developed two different parameterizations (ID#16 and 17) that use IWV and $T_s$ to derive ε and DLI. Global radiation transfer simulations were used by Dilley and O'Brien (1998) to determine the coefficients in the formulas.

## 6 Analysis and results

### 6.1 Evaluation of the existing formulas

The first objective of this analysis is to employ the observations carried out at the THAAO to evaluate the effectiveness of the 17 selected formulas in the form proposed by the authors (Table1), i.e., using the coefficients they determined.

The mean bias, RMSE, kurtosis, 5-th, 50-th, and 95-th percentiles of the difference $d_{i,j}$, KGE and $T_{skill}$ (see Section 4.2) obtained using 2017 THAAO data are shown in Figure 4. All calculated indices are reported in the Supplementary material, Tables S1

and S2, separately for years 2017 and 2018.The parameterizations capable of reproducing THAAO data are expected to produce low values of mean bias and RMSE, and large values of kurtosis, $T_{skill}$ (between 0 and 1), and KGE (between 0 and 1).

The Kling-Gupta efficiency presents a larger variability than $T_{skill}$, varying from 0.55 (Zhang, A, ID# 13 and Zhang, B, ID# 14) to 0.96 (Prata, ID# 12); on the other hand, $T_{skill}$ values are all larger than 0.9, with the exception of those obtained with

formulas ID# 4, 13, and14. The formulas producing the largest values of KGE and $T_{skill}$ are Jin (ID# 11), Ohmura (ID# 5), Dilley_B (ID# 17), and Prata (ID# 12); the largest values of the kurtosis are attained by Dilley_B and Prata.

The best performances in terms of bias and RMSE (absolute value of the bias < 6 W/m$^2$, and RMSE < 10 W/m$^2$) for both 2017 and 2018 are produced by the Dilley_B, Dilley_A, Jin, and Prata formulas (respectively ID# 16, 17, 11, and 12 in Figure 4).

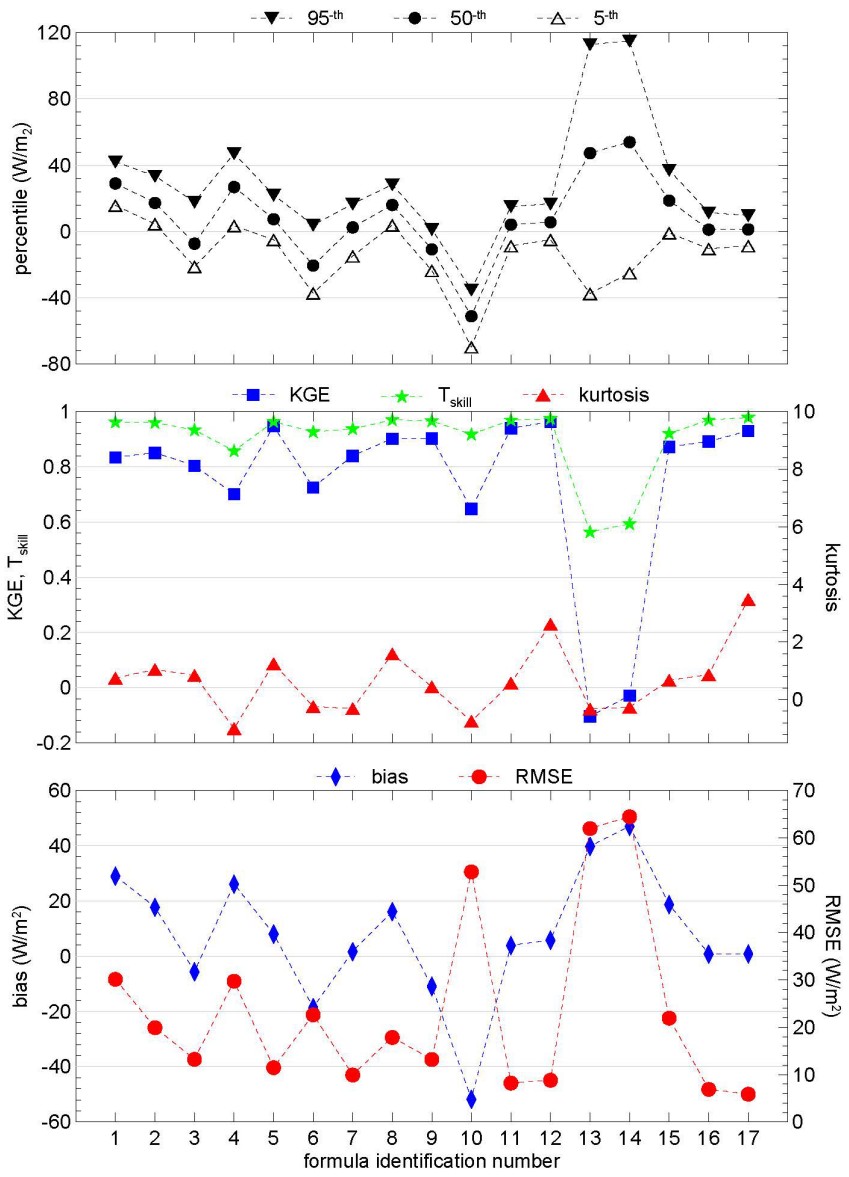

**Figure 4. Performance parameters of the different formulas for DLI listed in Table 1, tested using 2017 THAAO data: the upper panel shows 95-th, 50-th, and 5-th percentiles (filled triangles, filled circles and emptytriangles, respectively ) of the differences between estimated and measured values of DLI; the middle panel shows KGE (blue squares, left Y-axis), T$_{skill}$ (green stars, left Y-axis), and kurtosis (red triangles, right Y-axis); the bottom panel displays bias (blue solid diamonds, left Y-axis) and RMSE (red solid circles, right Y-axis). The formula corresponding to the different ID numbers is reported in Table 1.**



Thus, good performances are obtained by formulas that express ε using integrated water vapor and screen-level temperature (Dilley, ID# 16 and 17), IWV only (Prata, ID# 12), screen-level temperature and water vapour content (Jin, ID# 11), and screen-level temperature only (Swinbank, ID# 13).

Although the Dilley_B, Dilley_A, and Prata formulas (ID# 17, 16, and 12, respectively) and their coefficients were not specifically developed for the Arctic region, they achieve very good results using both IWV and $T_s$ (Dilley) or only IWV (Prata). These remarkable results suggest that these 3 parametrizations are less affected by the site-specific climatic conditions with respect to the other algorithms.The parametrization published by Jin (ID# 11) also shows very good performances and the lowest changes in terms of RMSE between the two years, confirming its effectiveness in the Arctic environmental

conditions for which it was developed.The formulas by Swinbank (ID# 3) and by Ohmura (ID# 5) produce the best results among those using only $T_s$.

The poor performance displayed by the Zhang's formula, which uses IWV, is probably due to the fact that it has been developed using data only from the melting season at Barrow and McGrath, and these data may not be appropriate to reproduce the annual changes of DLI in the Arctic. This is partially confirmed by the results of the Raddartz formula (ID# 15), that uses the same

dependence on IWV but obtains a better score.

Although the poor performances of Konzelmann's formula were unexpected (ID# 10), since it was developed and tested using measurements carried out on the Greenland Ice Sheet, the different conditions occurring over the ice sheet and at a coastal site, such as THAAO, may have played a role.

Small differences in the 2017 versus 2018 values of the indices calculated for each formula are found. Biases and $T_{skill}$ indices

of the same formula do not show significant differences between the 2017 and 2018 datasets, while RMSE is generally slightly smaller and kurtosis larger in 2018. This effect is due to the behaviour of the distribution tails, which are somewhat larger in 2017, as discussed in section 4.1.

In synthesis, formulas that use IWV and Ts perform somewhat better than the rest, although significant differences linked to the seasonal dependence of the dataset, or to specific conditions or particularly effective parameterizations exist.


## 6.2 Determination of THAAO-optimized coefficients

The second objective of this work is to find the coefficients of the considered formulas that best reproduce the THAAO datasets. To estimate the coefficients of the formulas using data measured at THAAO, the functions of the IDL® Software version 7.1.2 were used. Table 2 lists the functions used for each formula; details of the functions used can be found on the

web page https://www.nv5geospatialsoftware.com/docs/routines-1.html. All DLI formulas are derived by emissivity parameterizations except for ID#13, ID#16, and ID#17; with the exception of these cases, the emissivity was first calculated, and the performances were then obtained by comparing the measured and parameterized values of the DLI. Taking into account the size of the database, the functions were applied without considering any uncertainty in the measurements.

**Table 2. List of the function of IDL® Software used to derive the THAAO optimized coefficients.**

| ID# formula | IDL function |
|---|---|
| All ID# except those listed below | LADFIT |
| ID#11 | POLY_FIT |
| ID#16 and ID#17 | REGRESS |
| ID#7 and ID#12 | CURVEFIT |

The formulas with the coefficients optimised for the THAAO observations are reported in Table 3 for both year 2017 and 2018.

The analytical form of the Zhang_A (ID# 13), Zhang_B (ID#14) and Raddatz (ID# 15) formulas is the same, so the results obtained by optimizing those coefficients are displayed only for formula ID#13. Similarly, formulas ID# 8 and 9 of Table 1 have the same analytical form, and produce a single fit with respect to the THAAO data (ID# 8). The parametrization of Maykut, ID# 1, was implemented using both the annual mean (as the authors do) and the median values of $\varepsilon$. No significant differences in the coefficients and statistical results are found.

Figure 5 shows the performance of the parameterizations for 2017; in this case the same annual dataset has been used to determine the coefficients appearing in the formulas and the statistical indices discussed in section 4.2. Detailed statistical indices for years 2017 and 2018 are reported in the Supplementary material, Tables S3 and S4.

The bias is always smaller than 1.3 W/m$^2$, except for the Konzelmann formula (ID# 10) in both 2017 and 2018 and for the Zhang formula in 2018 (ID# 13 and). In general, similar values of the bias are obtained with data from 2017 and 2018, except

for the Zhang formula, for which a large change of the bias between the two years is found (-0.09 in 2017 and -2.01 W/m$^2$ in 2018).

The results of the Konzelmann formula (ID# 10) suggest that for the environmental condition of THAAO, the value of the exponent more suitable to express the dependence of $e_s/T_s$ is 1/7, and not 1/8.

In 2017 values of RMSE < 8 W/m$^2$ are found for the Dilley_B, Dilley_A, Prata, Jin, Idso, Satterlund, Ohmura, and Marshunova

(ID# 16, 17, 12, 11, 8, 7, 5, and 2, respectively) formulas; for all these cases the bias is < 0.25 W/m$^2$. The performances of the Ohmura and Marshunova formulas suggest that, when coefficients are retrieved using the same dataset, good results may be obtained also for formulas that include only dependencies on $T_s$ or $e_s$.

It is interesting to note that the values of RMSE in 2018 are generally lower than in 2017. This is possibly due to the meteorological conditions of 2017, which are characterized by a colder and drier winter and larger spread of the $T_s$, $e_s$ and

IWV values (see Section 4.1). The smallest RMSE values (<5.2 W/m$^2$ in 2018 and <6.0 W/m$^2$ in 2017) are attained by the parameterizations that include both the surface and the column information, i.e., the two formulas by Dilley and O'Brien (ID#16 and 17), and the one by Prata (ID# 12), for which $\varepsilon$ is a function of IWV.

**Table 3: Formulas from Table 1 with the coefficients determined using the data from THAAO. Data from 2017 and 2018 are used separately. Formulas 8 and 9, and formulas 13, 14 and 15, of Table 1 are equivalent when applied to the THAAO data. The same formula identification numbers of Table 1 have been used also in this Table.**

| ID# | Dataset year | Formula | Reference |
|---|---|---|---|
| 1 | 2017 | $\varepsilon = 0.6748$ | Maykut and Church, 1973 |
|   | 2018 | $\varepsilon = 0.6684$ | |
| 2 | 2017 | $\varepsilon = 0.6177 + 0.05 \cdot e_s^{0.04179}$ | Marshunova, 1966 |
|   | 2018 | $\varepsilon = 0.6182 + 0.05 \cdot e_s^{0.04004}$ | |
| 3 | 2017 | $\varepsilon = 9.701 \cdot 10^{-6} \cdot T_s^2$ | Swinbank, 1963 |
|   | 2018 | $\varepsilon = 9.802 \cdot 10^{-6} \cdot T_s^2$ | |
| 4 | 2017 | $\varepsilon = 1 - 0.3021 \cdot \exp[2.719 \cdot 10^{-4} (273-T_s)^2]$ | Idso and Jackson, 1969 |
|   | 2018 | $\varepsilon = 1 - 0.2990 \cdot \exp[3.311 \cdot 10^{-4} (273-T_s)^2]$ | |
| 5 | 2017 | $\varepsilon = 8.367 \cdot 10^{-3} \cdot T_s^{0.788}$ | Ohmura, 1981 |
|   | 2018 | $\varepsilon = 8.351 \cdot 10^{-3} \cdot T_s^{0.788}$ | |
| 6 | 2017 | $\varepsilon = 1.3836 \cdot (e_s/T_s)^{1/7}$ | Brutsaert, 1975 |
|   | 2018 | $\varepsilon = 1.4127 \cdot (e_s/T_s)^{1/7}$ | |
| 7 | 2017 | $\varepsilon = 1.040 \cdot [1 - \exp(-e_s^{T_s/2830})]$ | Satterlund, 1979 |
|   | 2018 | $\varepsilon = 1.042 \cdot [1 - \exp(-e_s^{T_s/2670})]$ | |
| 8 | 2017 | $\varepsilon = 0.6338 + 7.000 \cdot 10^{-5} \cdot e_s \cdot \exp(1500/T_s)$ | Idso, 1981; Andreas and Ackley, 1994 |
|   | 2018 | $\varepsilon = 0.6362 + 6.237 \cdot 10^{-5} \cdot e_s \cdot \exp(1500/T_s)$ | |
| 10 | 2017 | $\varepsilon = 0.4345 + 0.4565 \cdot (e_s/T_s)^{1/8}$ | Konzelmann et al., 1994 |
|   | 2018 | $\varepsilon = 0.4414 + 0.4408 \cdot (e_s/T_s)^{1/8}$ | |
| 11 | 2017 | $\varepsilon = [1.2953 - 0.008340 \cdot (T_s - 273.16) + 0.000144 \cdot (T_s - 273.16)^2] \cdot \left(\frac{e_s}{T_s}\right)^{1/7}$ | Jin et al., 2006 |
|   | 2018 | $\varepsilon = [1.2970 - 0.009604 \cdot (T_s - 273.16) + 0.000099 \cdot (T_s - 273.16)^2] \cdot \left(\frac{e_s}{T_s}\right)^{1/7}$ | |
| 12 | 2017 | $\varepsilon = 1 - (1 + IWV) \cdot \exp[-(0.6091 + 7.287 \cdot IWV)^{0.3305}]$ | Prata, 1996 |
|   | 2018 | $\varepsilon = 1 - (1 + IWV) \cdot \exp[-(0.5480 + 7.610 \cdot IWV)^{0.3225}]$ | |
| 13 | 2017 | $DLI = 134.02 + 50.916 \cdot \ln(IWV)$ | Zhang et al., 2001; Raddatz et al., 2013 |
|   | 2018 | $DLI = 128.42 + 51.251 \cdot \ln(IWV)$ | |
| 16 Dilley_A | 2017 | $\varepsilon = [1 - \exp(-1.66\,\tau)]$ with $\tau = 1.4951 - 1.1136 \cdot \left(T_s/273.16\right) + 0.7220 \cdot \left(\frac{IWV}{IWV_0}\right)^{0.5}$, and $IWV_0 = 25\ kg/m^2$ | Dilley and O'Brien, 1998 |
|   | 2018 | $\varepsilon = [1 - \exp(-1.66\,\tau)]$ with $\tau = 1.4502 - 1.0685 \cdot \left(T_s/273.16\right) + 0.7029 \cdot \left(\frac{IWV}{IWV_0}\right)^{0.5}$, and $IWV_0 = 25\ kg/m^2$ | |
| 17 Dilley_B | 2017 | $DLI = 52.083 + 112.403 \cdot \left(T_s/273.16\right)^6 + 117.532 \cdot \left(\frac{IWV}{IWV_0}\right)^{0.5}$, and $IWV_0 = 25\ kg/m^2$ | Dilley and O'Brien, 1998 |
|   | 2018 | $DLI = 49.885 + 117.098 \cdot \left(T_s/273.16\right)^6 + 109.674 \cdot \left(\frac{IWV}{IWV_0}\right)^{0.5}$, and $IWV_0 = 25\ kg/m^2$ | |

As occurred using the original parameters, the smallest year-to-year change of RMSE value occurs for the formulas by Jin (ID#11, RMSE of 6.58 and 6.96 W/m$^2$ respectively in 2017 and 2018), which also present very low values of the bias. The formula by Jin displays better statistical indices among those that use only surface information.

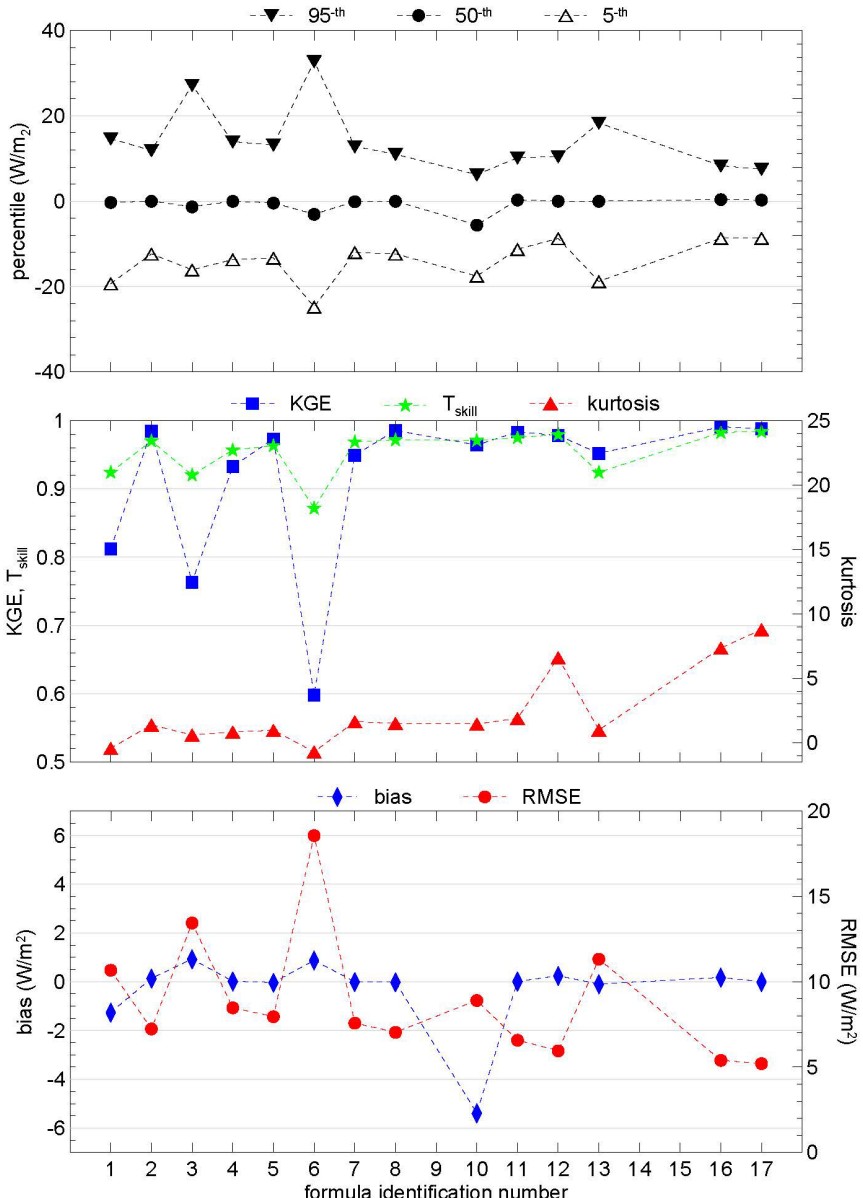

**Figure 5. Same as Figure 4, but the coefficients of the different parametrizations were derived from THAAO 2017 data and tested using the data of the same year. Please, note the change of vertical scales with respect to Figure 4.**

The Kling-Gupta efficiency and $T_{skill}$ show similar behaviour, with smaller values for the Maykut, Swinbank, Brutsaert, Zhang, and Raddatz (ID# 1, 3, 6, 13,). The poor performance of the formulas by Zhang and Raddatz, also with the optimization of the coefficients, suggests that a formulation of $\varepsilon$ as ln(IWV) is not adequate to describe the conditions occurring at THAAO. As expected, the optimization of the different formulas using data of both 2017 and 2018 produces a substantial improvement

of almost all statistical indices. Generally, the main effect of the optimization of the coefficients for the local conditions is to reduce significantly the value of the bias. A striking difference between the statistical indices calculated with the original coefficients found in the literature and those obtained with coefficients retrieved for the THAAO dataset can be noticed for the percentile distribution in Figures 5 and 6. Also, a general increase of KGE, $T_{skill}$, and kurtosis is found.

To test the representativeness of the formulas with respect to inter-annual variability, statistical indices were calculated using the annual data that were not used to derive the coefficients, i.e., data for 2018 were used for verification of the parametrizations obtained using coefficients derived employing 2017 data, and vice versa.

The main results of this analysis are shown in Figure 6. The full set of statistical indices for the two cases is reported in Tables S5 and S6 of the supplementary material.

Lowest values of the bias are produced by the Prata (ID# 12) and Jin (ID# 11) formulas. On the other hand, the best performances in terms of RMSE are obtained with the two formulas by Dilley and O'Brien (ID# 16 and 17). The Dilley_B formula (ID# 17) produces the smallest RMSE value. This parameterization shows a bias of about $\pm 1.2$ W/m$^2$.

The largest inter-annual variability of the bias is shown by the formulas that produce the worst results, and that hence appear less suited for representing the Pituffik environment.

The Jin's formula presents a very small bias for both years, and a nearly identical value of RMSE, confirming that it is poorly sensitive to changes of the dataset in the Arctic environment.

Testing the parametrization using different years produces changes in the values of the bias, but not in those of RMSE; this suggests that the value of the RMSE is mainly determined by the differences in meteorological conditions. The lower is the standard deviation, the better the parameterization uses the information to reproduce the DLI variability.

In most cases the values of the KGE and the $T_{skill}$ are larger than 0.95. High values of the kurtosis are attained, as also outlined for the previous analysis, by the Dilley_A, Dilley_B, and Prata formulas (ID# 16, 17, and 12).

These results suggest that, considering an annual variability of meteorological conditions similar to that of this study (see paragraph 4.1), the optimization carried out using one year of data can be applied to different years with an increase in in RMSE of $\sim$ 1-2 W/m$^2$, depending on the formula.


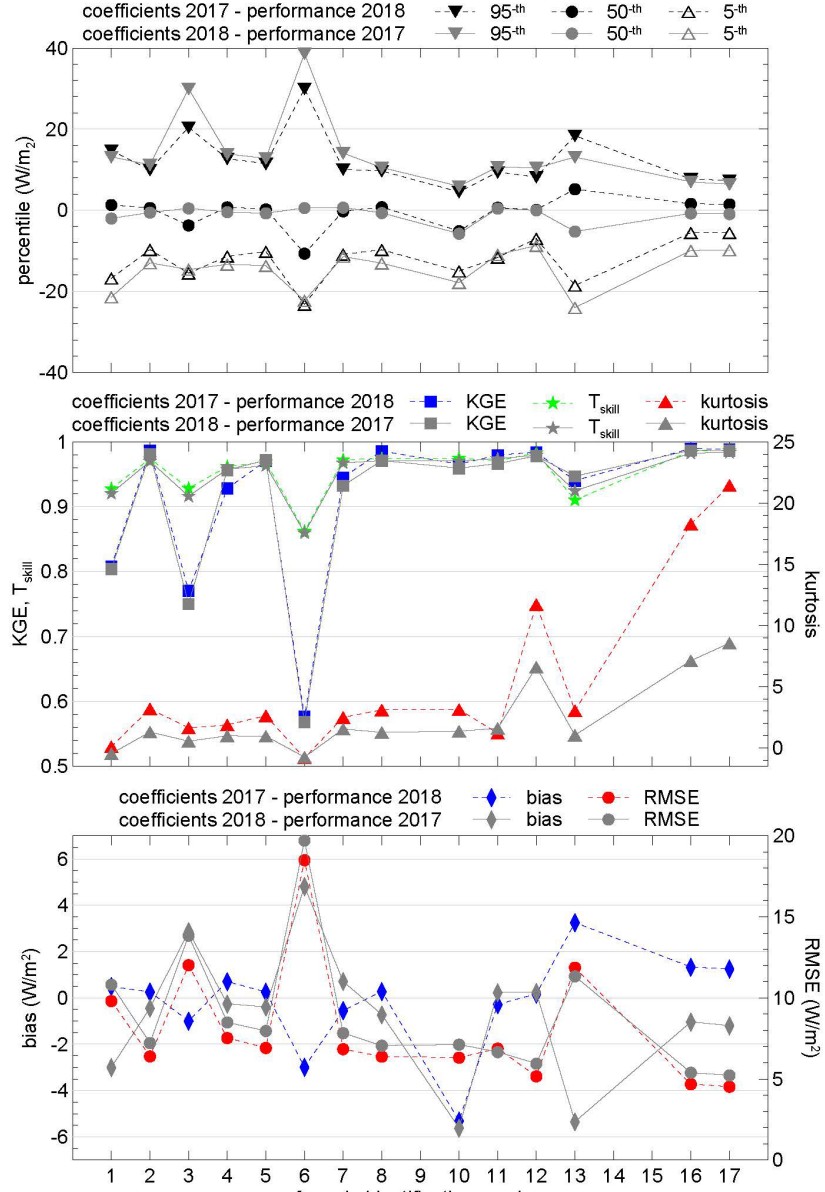

**Figure 6. Same as Figures 4 and 5, but this time the different parametrizations were optimized using one year of data and tested using the other (i.e., coefficients were determined with 2017 data and tested using 2018, and viceversa). This produces the two data points that are displayed for each formula. Grey symbols and lines are relative to coefficients derived from 2018 data, and performance parameters calculated with 2017 data. Coloured lines and symbols refer to results obtained using coefficients retrieved with 2017 data and performance parameters calculated with 2018 data.**


## 7 Summary and conclusions

This study investigates the performance of different formulas largely used in the literature to determine the clear sky downward longwave irradiance in the Arctic from meteorological parameters. In clear sky conditions the DLI is determined by the atmospheric concentration of greenhouse gases (mainly water vapor) in the lowest atmospheric layers and by their radiant temperature. The most reliable estimate of DLI is obtained by accurate measurements of the water vapor and temperature profiles, e.g. from radiosounding, used as input for radiative transfer models: however, this methodology is unpractical for large datasets, thus the need for DLI parameterizations.

The existing formulas need to be tuned to the environmental conditions of the application region because DLI estimate not from atmospheric profiles, but from proxies, such as screen-level measurements (e.g. air temperature or/and water vapour partial pressure) and/or integrated water vapor, were derived for a range of site-specific meteorological conditions. The effectiveness of each formula therefore depends on the representativeness of the parameters it uses to represent the atmospheric vertical profile and on its ability to simulate the emission processes in the lower layers of the atmosphere, which determine the DLI value. From this perspective formulas derived from the use of radiative transfer models are therefore expected to have more general validity than purely empirical ones.

The Arctic environment, in particular, poses specific challenges, due to the peculiar regime of temperatures, atmospheric water vapour content, and their vertical distribution.

The analysis has been carried out on the basis of high time resolution data collected during 2017 and 2018 at the Thule High Arctic Atmospheric Observatory (THAAO), that is located at 76.5°N, on to the North-Western Greenland coast. Measurements of all relevant parameters used in the existing formulas, including the integrated water vapour, are routinely carried out at THAAO, providing the possibility to test the algorithms.

A clear sky selection method was first developed to remove cloudy data. The cloud screening method is based on measurements of zenith sky brightness temperatures made in the 9.6-11.5 μm spectral window simultaneously to the other observations. Measurements in this spectral interval are poorly dependent on water vapour and very sensitive to clouds. The cloud screening uses a combination of thresholds on the zenith sky brightness temperature, its variability, and the persistence of low infrared brightness temperature values. The results of the clear sky selection have been verified with respect to sky imager pictures, and appears to be reliable and robust, even in the presence of thin cirrus clouds with optical thickness of about 0.08-0.1.

Seventeen different formulas have been chosen to be tested against the THAAO observations. Various statistical parameters have been adopted to assess the performance of the different formulas; these include bias, RMSE, the Kling-Gupta Efficiency, and the Taylor skill.

The analysis was carried out in two separate phases. In the first phase, the formulas with the originally derived coefficients have been tested against the clear sky THAAO dataset. In the second phase, the coefficients of all the considered formulas were optimized to Pituffik environmental conditions by deriving them from the THAAO dataset.

The availability of two full yearly cycles of data gives the possibility to: 1) test the formulas separately over two independent years, investigating the annual variability of the results, and 2) use two different datasets, one for the determination of the coefficients and another for the verification of the performance.

The main results of the analysis may be summarized taking into account the performance of the formulas with the original coefficient and those with the optimized coefficient. The coefficients of the parametrization optimized for the 2017 and 2018 meteorological conditions at the THAAO (see paragraph 4.1 Dataset characteristics) are provided in Table 3. The performance of the formulas and their applicability to other sites is mostly link to the variability of the meteorological condition used to determine the coefficients and the ability of the parameters to represent the water vapor and temperature vertical profile. In

general, themore similar the meteorological condition are to those of THAAO (see paragraph 4.1), the more the parameterizations found for THAAO can be directly applied with similar expected uncertainties (e.g. RMSE value). It is worth noticing that different combinations of water vapour and temperature profiles can provide the same DLI value.

Among the original formulas those that determine ε in terms of IWV and Ts, in particular those by Dilley and O'Brien (1998) (ID# 16 and 17), appear to perform better than those that are based on screen-level measurements only; in general, best

performances of formulas that use original coefficients produce biases $< 3$ W/m$^2$, and RMSE $< 7$ W/m$^2$. The formula by Jin et al. (2006) (ID# 11), which was developed on the basis of Arctic data using screen-level data, but that also takes into account the temperature and water vapor lapse rates, produces also very good results, and seems to be independent on the annual dataset used.

The optimization of the formulas, i.e., the determination of the coefficients based on the THAAO dataset, produces a significant

reduction of the bias and improvement of most statistical indices. Various formulas specifically developed for the Arctic appear to produce good results in matching THAAO data. There are however notable exceptions when the formulas have been derived from only a fraction of the year (e.g., Zhang et al., 2001), or over sites with very specific conditions (e.g., over the Greenland ice sheet, Konzelmann et al., 1994). Among all the re-tuned formulas, those using IWV and screen-level data produce better results; the mean bias, estimated with a separate dataset from the one used to determine the coefficients, is $< 1.33$ W/m$^2$ for

the formulas by Dilley and O'Brien (ID#16 and 17), and Prata (ID# 12). The same formulas present the minimum values of RMSE, $\leq 6$ W/m$^2$, that are associated with maxima of the kurtosis and high T$_{skill}$.

The analyses carried out on two different meteorological years indicate that the optimization of a formula carried out on one year is applicable to different years producing an increase in RMSE of ~1-2 W/m$^2$ depending on the selected formula; these results are strictly valid for the database used. This analysis suggests that for other sites the formula optimised on a reference

year can be applied to other years with similar variability of atmospheric conditions, with an expected small increase in RMSE. Considering both the original and optimized formulas in terms of RMSE the parametrization of Dilley and O'Brien (ID#16 and 17), Prata (ID# 12) and Jin (ID# 11) present the best performance on the used datasets. It is worth noticing that at the THAAO, expressions by Dilley and O'Brien (ID#16 and 17) and by Prata (ID# 12), which were developed on the basis of global data and radiation transfer model simulations, appear to perform better than formulas specifically developed for Arctic

conditions, even when the former are applied using their original coefficients. This is probably due both to the use of multiple

parameters to estimate the DLI and the formulation derived from radiative transfer simulations that better expresses the relationship between the input atmospheric parameters and the DLI. On the other hand, formulas expressing ε only as a linear function of ln(IWV) appear to produce unsatisfactory results when applied to the THAAO database.

Thus, specialized formulas allow to retrieve the clear sky DLI within about 5-7 W/m$^2$, as also suggested by the distribution of the percentiles, that is of the same magnitude of the uncertainty of the DLI measurements carried out by high quality pyrgeometers. We intend to use these estimates to derive the IR cloud radiative perturbation that is calculated as the difference between the measured DLI in cloudy conditions and the corresponding clear sky DLI. Uncertainties on the estimates of clear sky DLI directly influence our capability to determine the cloud radiative perturbation, which is fundamental to assess the role that clouds play in the Arctic climate. There is the need to reduce the uncertainties on these determinations, since a relatively large uncertainty on the estimated values of the clear sky DLI impairs our ability to determine the radiative effect of thin and even moderate clouds.

*Data availability.*

The time series of the data can be visualized and downloaded through the THAAO web site (https://www.thuleatmos-it.it/data/index.php).

Meloni, D., Di Sarra, A., Di Iorio, T., Pace, G., Muscari, G., Iaccarino, A., and Calì Quaglia, F.: Downward Shortwave Irradiance at the Thule High Arctic Atmospheric Observatory (THAAO_DSI) [Data set]. Agenzia Nazionale per le nuove tecnologie, l'energia e lo sviluppo economico sostenibile (ENEA), https://doi.org/10.13127/THAAO/DSI, 2022.

Muscari G., Di Sarra A., Di Iorio T., Pace G., Meloni D., Sensale G., Calì Quaglia, F., and Iaccarino A.: Meteorological data at the Thule High Arctic Atmospheric Observatory (THAAO_Met). Agenzia Nazionale per le nuove tecnologie, l'energia e lo sviluppo economico sostenibile (ENEA), https://doi.org/10.13127/THAAO/MET, 2018.

Pace, G., Muscari, G., di Sarra, A., Calì Quaglia, F., Meloni, D., Iaccarino, A., and Di Iorio, T.: Infrared Brightness Temperature at the Thule High Arctic Atmospheric Observatory (THAAO_IBT) [Data set]. Agenzia Nazionale per le nuove tecnologie, l'energia e lo sviluppo economico sostenibile (ENEA), https://doi.org/10.12910/DATASET2023-001, 2023.

Pace, G., Muscari, G., di Sarra, A., Calì Quaglia, F., Meloni, D., Iaccarino, A., and Di Iorio, T.: Integrated Water Vapor measured by an HATPRO microwave radiometer at the Thule High Arctic Atmospheric Observatory (THAAO_IWV_HATPRO) [Data set]. Agenzia Nazionale per le nuove tecnologie, l'energia e lo sviluppo economico sostenibile (ENEA), https://doi.org/10.12910/DATASET2023-002, 2023.

*Authors contributions.* Conceptualization, G.P.; Data acquisition: G.M. performed the surface meteorological measurements; D.M. performed the measurements of DLI; G.P. performed the measurements of zenith sky IR radiance and IWV; Methodology, G.P.; Formal analysis, G.P., D.M.; Editing-original draft preparation, G.P., A.d.S.; Writing-review and editing,

all authors; Funding acquisition, G.M., D.M., G.P., V.C. All authors have read and agreed to the published version of the manuscript.

*Competing interests.* The authors declare that they have no conflict of interest.

*Acknowledgements.* This research has been funded by the Italian Ministry for University and Research through the following Projects: Study of the water Vapour in the polar AtmosPhere (SVAAP), Observations of the Arctic Stratosphere In Support of YOPP (OASIS-YOPP), and CLouds And Radiation in the Arctic and Antarctica, (CLARA$^2$), all supported by the Italian Antarctic Programme; and Effects of changing albedo and precipitation on the Arctic climate (ECAPAC), supported by the Italian Arctic Programme; national activity of the Aerosols, Clouds and Trace Gases Reasearch Infrastructure (ACTRIS-IT). This research has also been supported by the MACMAP project funded by Istituto Nazionale di Geofisica e Vulcanologia. This is a contribution to the Year of Polar Prediction (YOPP), a flagship activity of the Polar Prediction Project (PPP), initiated by the World Weather Research Programme (WWRP) of the World Meteorological Organisation (WMO). We acknowledge the WMO WWRP for its role in coordinating this international research activity.

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
