# Peer review of "Verification of parameterizations for clear sky downwelling longwave irradiance in the Arctic"

_Atmospheric Measurement Techniques, 2023_

## Author Comment (AC1)

*Verification of parameterizations for clear sky downwelling longwave irradiance* by Pace et al. uses data collected from Thule, Greenland, to evaluate parameterizations for estimating the downwelling longwave flux at the surface based on screen height meteorological measurements. I have not seen an intercomparison quite like this and I think the results are of interest and publishable in *AMT*. I have a few minor comments for the authors to consider before this manuscript is published.

Dear Dr Cox, we really appreciate your useful comments that focus on and clarify some aspects of the paper. In the following the answers to individual comments:

- A major revision that expands this analysis to include the other YOPP supersites (https://www.polarprediction.net/key-yopp-activities/yoppsitemip/the-yopp-arctic-and-antarctic-supersites/) would make the conclusions more broadly interpretable. However, I respect the scope that the authors are setting and if they do not wish to expand the analysis, I think it would be beneficial to provide some additional text contextualizing the Thule area. For example, northwest Greenland is within a small sub-region of the Arctic where the atmosphere is generally drier (lower IWV) (e.g., Cox et al. 2012) and indeed in the vicinity of 1 cm IWV, there are spectral effects (Cox et al. 2015) that I suspect could impact derivation of coefficients for the parameterizations (more challenging still if one were to interpret Thule data as representative of higher elevations over the ice sheet). Clouds at Eureka, Canada, also in this dry region, are higher, colder, and thinner (e.g., Shupe et al. 2011) compared to much of the rest of the Arctic. Thus, Thule may not be an ideal analogue for either the ice sheet or for the Arctic as a whole. That does not to devalue the results presented here, but is relevant context for readers to understand.

We agree that extending this work to other sites could be useful to people involved in surface radiation budget research in the Arctic, both in the context of the YOPP and in a wider context, given that there are only few detailed works for estimating the DLI in this region. On the other hand, extending the analysis to other sites is not automatic as it would require an in-depth check of the different databases, also thinking about the methodology for determining clear skies, which goes beyond the scope of this work.
We also agree that more details to contextualize the characteristics of Pituffik area (formerly known as Thule) should be provided. We believe it is important to characterize at the best the site in terms of meteorological conditions when carrying out this type of study, so that a possible reader interested in using the results of the paper can do so with knowledge of the facts and it is precisely for this reason that the distributions of the values of $e_s$, IWV and $T_s$ were shown in the manuscript, implicitly indicating the limits of applicability of the results.

Following your suggestion, a description of the characteristics of THAAO has been added at line 83.
*The THAAO is located on South Mountain, at 220 m a.s.l., near the Pituffik Space Base (formerly known as Thule Air Base), along the north-western coast of Greenland at about 3 km from the sea and 11 km from the Greenland ice sheet (GrIS). Therefore, the THAAO environment is typical of the northern coastal area of Greenland, i.e., influenced by both the GrIS which generates strong katabatic winds, and by the sea, especially in summer when open waters prevail over sea ice. Pituffik is also located in a region, which includes the area northwest of Greenland and the Ellesmere Island, characterized by an atmosphere particularly dry (Cox et al. 2012), with higher, colder and thinner clouds with respect to what is found in other areas of the Arctic (Shupe et al. 2011).*

- Long and Turner (2008) is essential reading on the topic presented here and should be referenced and considered for this study as well as future work (as indicated at L432-438).

We agree with the reviewer that it was a lack not to cite the work of Long and Turner (2008). Unlike the other parameterizations used in this work, the one presented by Long and Turner (2008) uses variable coefficients not only as a function of physical parameters, but also includes the effort to explicitly represent the daily variability of the parameters in the studied sites. This parameterization therefore requires a more specific approach which involves the optimization of the so-called "Lapse Rate Coefficient" (a coefficient that depends on the lapse rate originally defined by Brutsaert, 1976) by evaluating its variations before sunset, during the night and after dawn, and interpolating in time the obtained results. Given the different approach with those presented in our manuscript, we chose not to test this parameterization which requires more study and introduces site-specific daily variability. Similarly, we decided not to

include the methodology presented by Dürr and Philipona (2004), who optimize the Lapse Rate Coefficient by taking into account the periodic annual and daily variability of the studied sites.

On the other hand, the parameterization presented by Jin et al. (2006) is also based on the concept of the Lapse Rate Coefficient, and it is expressed as a function of the $T_s$ only, derived by an analysis of more than 700 radiosoundings launched from the Arctic station of Resolute Bay. This parameterization has been used and discussed extensively in this work.

Accordingly, the two papers by Long and Turner (2008) and Dürr and Philipona (2004) have been included in the manuscript, providing a brief description of their approach in line 53.

*The parameterizations in Dürr and Philipona (2004) and Long and Turner (2008) differ from those considered in this work because they use explicit dependences on the annual and daily variability of the observed atmospheric parameters and DLI at the measurement site and therefore require specific analyses. Both the works improve the parameterization of atmospheric emissivity presented by Brutsaert (1975) by refining the estimation of the so-called Lapse Rate Coefficient. Dürr and Philipona (2004) approximated the diurnal and annual cycle of the considered sites using a periodical function, while Long and Turner (2008) analyzed separately the daytime and nighttime behavior of the Lapse Rate Coefficient interpolating the daily results during sunset and sunrise; they also applied this method to the Arctic site of North Slope in Alaska, finding differences within ± 4 W/m² between the measured and observed DLI values in 68% of cases.*

- Could you explain in more detail (more quantitatively) the accuracy of your clear-sky detection method (L67-69)? Could you clarify if it is necessary to capture all instances of clear-sky or (I think) only to capture a large sample of confidently detected clear-sky? Could you clarify the sensitivity of the vulnerability in this method to assigning "clear-sky" to cases with high, cold, optically-thin (e.g. cirrus) clouds?

The choice to use the pyrometer instead of the pyrgeometer itself to determine clear sky conditions, presents the advantage of a larger sensitivity to the presence of thin clouds, but the disadvantage of a reduced portion of sky detected at the zenith; this is the main factor that makes it difficult to associate a quantitative accuracy to the developed clear-sky detection method.

Simulations of the pyrometer zenith brightness temperature (IBT) and of the DLI have been carried out using the MODTRAN5.3 radiative transfer model, to evaluate the uncertainty associated with the presence of thin cirrus clouds. The aim was not only to verify the sensitivity of the pyrometer to the presence of cirrus, but also to quantitatively determine the influence of these clouds on the DLI.

The main characteristics of the simulations are summarized in the following table.

| Atmopheric profile | Subarctic winter | Subarctic summer |
|---|---|---|
| IWV | 0.3 cm | 1.2 cm and 1.5 cm |
| Acloud base altitude | 8 km | 8 km |
| cloud base temperature | 220.6 K | 239.2 K |
| Geometrical depth | 1 km | 1 km |
| Cloud type | cirrus | cirrus |

The MODTRAN internal cirrus model, called cirrus standard model, is based on ice particles with 64 μm effective radius. Overcast conditions are assumed. The cirrus optical thickness values have been of respectively 0.03, 0.1, 0.3, 1, 2, 3, 5. The values of 0.03, 0.3 and 3 has been chosen considering the pioneering work of Sassen and Cho (1992), who defined these thresholds to define sub-visible, thin and opaque cirrus clouds. A winter and two summer cases has been simulated, whereby 0.5 cm and 1.2 cm are the average seasonal values, and 1.5 cm is used to assess the sensitivity of IBT and DLI to larger column water vapor below the cloud.

The increase in the value of DLI and IBT as a function of the cirrus optical thickness is presented in Figure1a for both the winter and summer case. The labels close to the lines show the increase in the value of DLI and IBT compared to that of the clear sky simulation.

[Figure]

*Figure 1a. The plots show the DLI (in the bottom panel) and the IBT (in the top panel) increase as the optical thickness of the cirrus cloud increases, for the winter (left) and the summer (right) case respectively. The labels show the increase in the value of DLI and IBT compared to that of the simulation in the absence of cirrus. A horizontal dotted line highlights the temperature of the cloud base in the winter and summer cases.*

The clear-sky IBT for the winter case is 158 K, which is below the pyrometer's calibration range, i.e. down to 173 K, but within its measurement range, i.e. down to 123 K. Even the presence of cirrus with optical thicknesses of 0.03 and 0.1 determines an increase in the pyrometer signal of 4.3 and 11.3 K, respectively equal to an increase in IBT of 2.7% and 7.1%, respectively, compared to cloud-free conditions. Both these values are clearly visible compared to the background signal and within the pyrometer measurement uncertainty that, considering the temperature at the base of the cirrus, is approximately ±1.6/2.0 K. On the other hand, the increase in DLI with respect to its cloud-free value (i.e. 168 W/m²) is just 0.8 (0.5%) and 2.7 (1.6%) W/m², that is particularly small also taking in to account the uncertainty on DLI measurements that are estimated to be ±5 W/m².

Similar results are found for the summer simulations. In this case, two simulations were performed taking into account different values of IWV. For IWV equal to 1.2 cm, even considering cirrus optical thickness of just of 0.03 and 0.1 the IBT increases, compared to that with clear sky (i.e. 191.1K), by 1.6 (0.86 %) and 5.0 (2.66%) K, respectively. It should be kept in mind (see paragraph *2 Site and measurement* of the manuscript) that, by decreasing the temperature difference between the cloud base and the pyrometer, the IBT associate uncertainly (±1.3 K) decreases. The same cirrus cloud determine an increase in the clear skies value of DLI, i.e. 279.5 W/m², respectively of 0.8 (0.3%) and 2.7 (0.97%) W/m² for 0.03 and 0.1 optical depth. Further increasing in the IWV does not substantially change the results, although the decrease of the IBT sensitivity to this cirrus is larger than that of DLI.

In summary, the simulations highlight the larger sensitivity to the presence of thin clouds of the pyrometer compared to the pyrgeometer, especially in the polar environment characterized by low IWV and therefore larger transparency in the atmospheric window around 10 μm.
However, our method for defining clear skies is based more on the variability of the signal than on its intensity.
These simulations highlight how the better sensitivity of the pyrometer induces a larger variability of the signal which is therefore more suitable than that of the pyrgeometer to be used to define clear sky conditions, also because it is much less influenced by the IWV changes (see Figure 2a).
Furthermore, it must be considered that zenith measurements are generally more sensitive to the 2D spatial variability of the cloud than hemispheric measurements. It is therefore unlikely that the algorithm can indicate a *zenith clear sky case* condition corresponding to the presence of a cloud that significantly influences the DLI.

Considering the results of the simulations and the visual inspection of the data (e.g. Figure 2a and discussion in the following), using a conservative approach, the methodology correctly evaluates the presence of cirrus clouds with an optical thickness larger than approximately 0.07-0.1 that, depending by the atmospheric profile (mostly IWV), and the

physical (cloud base height, geometric thickness) and microphysical characteristics (ice content, size, shape...) of cirrus should determine an increase in the IBT value of no less than 7/15 K.

Considering the interest that both reviewers have shown in the clear sky algorithm and to better answer their questions, we present and discuss an example of how the algorithm operates to recognize clear sky cases, or perhaps it would be better to say cases where the presence of clouds does not appreciably influence the DLI.
Figure 2a shows six days of IBT, DLI, IWV and $T_s$ measurements collected in February 2018; cases recognized as clear sky are shown as green points.

[Figure]

*Figure 2a. From bottom to the top: time series of DLI, IBT, IWV and Ts for the period from day number 43 to 49 of 2018, i.e. 12-18 February. As expected, the behavior of DLI is strongly dependent by the variability of IWV and Ts, which do not always show similar patterns.*

The algorithm proves to have enough sensitivity to detect very thin clouds that only slightly increase DLI, but determine a IBT increase of approximately 15 K or less (e.g., see the IBT on the beginning of day 43). Although the algorithm proves to work well in relation to the purpose of this analysis, we have chosen to present this case to highlight what has already been mentioned in article lines 167-175, i.e. the importance of visual control of the clear sky.
In the time interval from 46.55 to 46.7 the IBT does not highlight any clouds, while the DLI shows a decrease suggesting residual coverage of the sky. Although this hypothesis cannot be discarded, observation of the sky images shown in Figure 3a suggests another explanation.
Just as happened on the dome of the sky imager between 6 and 18, it is probable that some frost condensed on the dome of the pyrgeometer and then slowly sublimated. This can occasionally occur even if the pyrgeometer is ventilated. In addition, the BSRN quality tests on DLI may not detect such effect. It should be remembered here that the surface of the window of the pyrometer is ventilated with an air flow coming from inside the observatory and is therefore less subject to these phenomena.
These phenomena and the presence of snowfall were the main obstacle to the correct and automatic functioning of the algorithm. Thanks to a visual analysis of the entire dataset their impact is considered negligible.

[Figure]

| | | | |
|---|---|---|---|
| 00:47 | 2:00 | 4:00 | 6:00 |
| 8:00 | 10:00 | 12:00 | 14:00 |
| 16:00 | 18:00 | 20:00 | 22:00 |

*Figure 3a. Bi-hourly sky images of day 46 of 2018 (15 February 2018).*

Also taking into account these effects it is much more complex to quantitatively demonstrate that the passage from a *zenith clear sky case* condition to a *clear sky*, i.e. hemispheric clear sky, does not include the residual presence of clouds. The choice to consider a relatively long series of *zenith clear sky cases* to define a *clear sky* condition (please remember that our method identifies a clear sky condition, e.g., at 12:00 if on the 61 IBT measurements ranging from 11:30 to 12.30 at least 45 zenith clear sky cases occur) is based both on the visual analysis of the dataset and the conclusion of the interesting work by Kassianov et al. (2004) that states "for a relatively short averaging time (15 min), the zenith-pointing observations with a narrow FOV (lidar/radar) can greatly (more than 100%) overestimate/underestimate the cloud fraction".

Summarizing the above to directly answer the reviewer's question:
Could you clarify if it is necessary to capture all instances of clear-sky or (I think) only to capture a large sample of confidently detected clear-sky?
The choice to define a clear sky measurement by evaluating 61 *zenith clear sky measurements* was maybe a little bit conservative, but (also taking into account the large dataset used) it was preferred to remove some clear sky data rather than include dubious data.

**Reference**

Sassen, K. and Cho , B.Y. :Subvisual-thin cirrus lidar data set for satellite verification and climatological research, *J. Appl. Meterorol.*, 31, 1275–1285, 1992.

Kassianov, E., C. N. Long, and M. Ovtchinnikov (2004), Cloud sky cover versus cloud fraction: Whole-sky simulations and observations, *J. Appl. Meteorol.*, **44**, 86–98.

**Some editorial comments:**

- L37: Ohmura et al. (2001) is a good reference here as well.

The reference has been added to the sentence.

- Figure 1: I assume the straight lines in 3 & 4 (just before 2018.75) and the bottom panel (between 2017.75 and 2018.00) are artifacts of the plotting technique? Some other short periods in year 2 bottom panel as well. Can these be removed so we can properly visualize data gaps?

Done

- The word "which" is used frequently in places where the correct word is "that" (e.g., L16, 24, 25 but then throughout the text).

The correction has been implemented throughout the text.

---

## Author Comment (AC2)

**RC2**: ['Comment on amt-2023-181'](), Claudia Di Biagio, 07 Oct 2023
First of all I would like to sincerely apologize for the delay of my revision.

The paper by Pace et al. investigates the performances of different parameterizations to estimate the clear–sky downward longwave irradiance (DLI) based on comparison against ground–based observations at the high latitude station of THAAO, Greenland. A set of 17 empirical formulations for the DLI are tested. The authors use two full year observations (2017 and 2018) with the aim of: 1/ evaluate original formulas against 2017 data; 2/ use 2017 data to provide optimized coefficients for the THAAO conditions; 3/ test the optimized coefficients against 2018 data. Observations at THAAO used in the present analysis include IBT (infrared zenith sky brightness temperature) and DLI, as obtained from pyrgeometer and pyrometer instruments, and meteorological parameters. The pyrometer data are used to derive cloud–free periods based on an original algorithm, while the pyrgeometer data are used as direct comparison against calculated DLI with the 17 different formulations.

The paper is well written, clear and logically organized. The dataset, the procedure, the figures, and the results are clearly presented and discussed. The final scope of this work, that is providing optimized estimates of the clear–sky DLI to use then in future works to estimate the cloud direct radiative effect in the infrared spectrum at THAAO, is clearly identified and of potential interest also for other researcher. The work fits within the scope of AMT and publication can be suitable after minor revisions. I propose here some suggestions to potentially clarify some aspects and broad the conclusions, as well as some minor technical remarks.

We thank Dr Di Biagio for the useful comments and suggestions that improve the clarity and effectiveness of the work done and its presentation.

Abstract, line 20: "The bias displays a significant improvement when the coefficients of the different formulas are calculated using the THAAO dataset." This is somehow obvious, I guess. I would suggest: "As expected, the bias displays a significant improvement when the coefficients of the different formulas are calculated using the THAAO dataset."

Done

Abstract, line 21–22: "The presence of two full years of data allows the investigation of the inter–annual variability,". I am not sure that two years of data are sufficient for performing an analysis of the inter–annual variability. On the contrary, as discussed in Sect. 4.1, the two years are interesting to be used in this study since they show different conditions to be tested. I would put this aspect in evidence in the abstract.

The sentence "The presence of two full years of data allows the investigation of the inter-annual variability, and the use of different years for the determination of the coefficients and the evaluation of results" has been changed into "*The presence of two full years of data allows the determination and the applicability of the coefficients for singular years and the evaluation of results*".

Introduction, lines 28–29: references would be good at the end of this first sentence

The following reference has been added to the sentence.

Taylor, P. C., Boeke, R. C. , Boisvert, L. N. , Feldl, N. , Henry, M. , Huang, Y., Langen, P. L., Liu, W., Pithan, F., Sejas, S. A. and Tan, I.: Process Drivers, Inter-Model Spread, and the Path Forward: A Review of Amplified Arctic Warming, Front. Earth Sci., 9, doi: 10.3389/feart.2021.758361, 2022.

Introduction, line 40 and wherever this applies: very minor comment, but when possible it is better avoiding to start sentences with acronyms

Done

Introduction, lines 40–42: references would be good at the end of this sentence

The following reference has been added to the sentence.

Shupe, M. D. and Intrieri, J. M.: Cloud Radiative Forcing of the Arctic Surface: The Influence of Cloud Properties, Surface Albedo, and Solar Zenith Angle, J. Climate, 17, 616–628, https://doi.org/10.1175/1520-0442(2004)017<0616:CRFOTA>2.0.CO;2, 2004.

Section 3: Two comments here:

1/ it would be useful to show (even in the supplementary) a plot illustrating the distribution of clear–sky periods during the two years under consideration. Somehow the discussion of the results in terms of seasonality is missing and it would be good to start from the identification of the distribution of clear sky periods.

We agree with the reviewer that a more detailed description of the annual data distribution could add useful information to the work, but we believe that including this other information in the paper could make the work too long. Moreover, from our perspective, it is not completely within the scope of the article.
Therefore we have prepared additional material presenting both the annual variability of the data and an evaluation of the performance of the parameterizations in winter and summer by applying the formulas in the months of January-February and July-August of 2017 and 2018.

In our opinion, a discussion on the performances of the different parameterizations as a function of seasonality does not fall within the main objectives of the work. Indeed, the purpose of the study is to find parameterizations able to reproduce the DLI starting from the variability of the parameters that most influence it, i.e. $T_s$, $e_s$ and IWV.
The main goal of this work is to evaluate the effectiveness of these parameterizations throughout the year in the Arctic sites similar to THAAO, i.e. for all those stations that present a distribution of the values of these parameters similar to that reported in Figure 3 of the paper, analyzing which parameterizations best reproduce the DLI values depending on the atmospheric variable used, providing the most suitable coefficients for the purpose and indicating an uncertainty to be associated with the parameterizations used regardless of the year or period of study. Also the comparison between the results found in the two years have this meaning.

We agree with the reviewer that using "only" two years the term "interannual variability" can be misunderstood. On the other hand, we believe that one of the most interesting points of the work is using two different annual databases, obtaining the optimized coefficients for one year and applying them to the other year, verifying that the uncertainty associated with the parameterization (i.e. RMSE value) does not present substantial variations, even if the two years show differences in the evolutions of the atmospheric variables.
In this way, the applicability of the results obtained if applied to different years as regards for THAAO or for other sites that present a meteorological variability comparable to that reported in Figure 3 was verified (at least on two full annual cycle).
The functions used to optimize the parameterizations (see the specific answers to the reviewer comments in the following) tend to minimize the bias between the measured and the parametrized DLI values, obviously depending on the dataset used. For example, in the hypothesis of a field campaign of limited duration, better results would probably have been obtained by optimizing the parameterizations over that time interval (or rather, over an slightly longer interval). Evaluating the monthly or seasonal performance using an optimized annual parameterization has the meaning of evaluating how much the uncertainties associated with a given

parameterization remain constant during the year (for example, in terms of RMSE), without however changing the meaning of the annual results.

Following a different approach to this work, it would have been possible to group the dataset used at a seasonal level and then carry out an analysis similar to the one carried out. By dividing the data in sub-annual periods, like seasons, we would have found performances that were probably slightly better than the annual ones.

On the other hand, the results thus obtained would have presented some limitations including:
1) the difficulty of having several uncertainties to apply during the year.
2) the need to manage the overlapping periods of the different seasons in terms of choosing the most suitable set of coefficients for each parameterization and different meteorological condition.
3) a reduced applicability of the results to other sites.

At the end of the answer to reviewer seasonal variability of cloud free occurrence is presented, together with the performance of the annual parameterizations for the months of January-February and July-August 2017 and 2018, respectively based on the parameterizations optimized on the annual data of 2017 and 2018, which will be also provided as a single attachment (supplementary material).

2/ I understand this cloud–screening procedure is somewhat original. If this is correct, would not be worth to put this more in evidence in the paper? This could imply a few changes in the title/abstract/general discussion. I let the authors to evaluate if this is pertinent or not.

We acknowledge the reviewer for the suggestion. Given the interest also shown by the other reviewer, we are evaluating the hypothesis of dedicating a specific work to the characterization of this cloud screening method which, however, goes beyond the scope of the present work. In fact, to better refine this method it would be necessary to at least add data from a ceilometer and a sky imager in order to have (at least for daytime measurements) independent measurements of the presence of clouds and cloud cover to validate the cloud screening method.

Section 6, general comment: the analysis of the seasonal dependence of the performances of the different formulas is not provided. Have the authors' analysed possible seasonal changes in the performances of the different formulas both for original and the THAAO–optimized formulations? Can they provide some insight on this aspect or derive useful information from this analysis? In some points the discussion the authors mention that original formulations derived based on partial year data are not good because of this limited time period used as reference. In order to better understand this point, it would be appropriate to discuss the seasonal coverage of clear–sky data used for THAAO analysis (see previous comment) and the seasonal changes (if there are) of the performances of the different formulas.

See the answer to the previous comment and the new supplementary material.

Section 6.2: can more details be provided on the procedure to estimate the optimized coefficients?

To estimate the coefficients of the different parameterizations starting from the data measured at the THAAO Observatory, the routines of the IDL software version 7.1.2 were used. As shown in Table 2 of the manuscript, all DLI parameterizations are based on the emissivity parameterization except for ID#13, ID#16, and ID#17; therefore, with the exception of these cases, the emissivity was calculated, and the performances were subsequently obtained by comparing the measured and parameterized values of the DLI.
Most formulations can be linearized as a function of the parameters $e_s$, $T_s$, IWV and/or a combination of them, e.g. $e_s/T_s$; in this case the LADFIT function, which uses a minimum absolute deviation method was applied. In other cases, such as for ID#11, the emissivity was calculated using the POLY_FIT function which applies a polynomial least squares fit. For the parameterization of ID#16 and ID#17 the REGRESS function to

perform a multiple linear regression fit has been used. For the parameterization of ID#7 and ID#12 we applied the CURVEFIT function which uses a gradient expansion algorithm to compute a nonlinear least squares fit. Given the size of the database, the fits were applied without considering any uncertainty in the measurements. Specific details of the used functions can be found on the web page https://www.nv5geospatialsoftware.com/docs/routines-1.html.

Section 7, lines 400–401, I would skip those two lines, it is a repetition

The sentence has been modified to cite the information regarding the number of parameterizations used in the article in the conclusions (now line 421).
It now reads: *"Seventeen different formulas have been chosen to be tested against the THAAO observations"*

Section 7, line 408: see my previous comment on the "inter–annual variability"

The sentence has been modified accordingly.

Section 7, line 432: worth to point out that the 5–7 Wm$^{-2}$ is within the uncertainty of the DLI by pyrgeometer, as discussed earlier in the paper

The sentence at line 456 has been changed into "*Thus, specialized formulas allow to retrieve the clear sky DLI within about 5-7 W/m$^2$, as also suggested by the distribution of the percentiles, that is of the same magnitude of the uncertainty of the DLI measurements carried out by high quality pyrgeometers.*"

Section 7, general comments: based on the analysis and conclusions of the paper, two points can be raised and maybe deserve a word of conclusion from the authors. Firstly, as at the end the DLI parameterizations are site–dependent, one may ask if, for a specific site, the tuned–parameters derived for a specific year can be used to predict the DLI no matter which year of application or not. In other words, a discussion about which statistics/time range can be reasonable to consider for proper use of tuned–parameterizations should be added. Secondly, does the authors have any hypothesis on why the DLI parameterizations are site–dependent or which parameters seems to affect the most the formulation?

As briefly mentioned in the introduction, line 36-40, the DLI is determined "by the atmospheric concentration of the main greenhouse gases and their radiant temperature". The best estimate of the DLI should be obtained by knowing the vertical distribution of the main greenhouse gases, mainly water vapor, and of the air temperature and using an updated radiative transfer model.
DLI parameterizations are as accurate as they are able to represent the atmospheric profiles, mainly in the lowest layers. Their adaptability to different sites depends mainly on the fact that each site presents different meteorological conditions, not only in terms of variability at screen level, but also in the vertical profile (e.g. temperature inversions).
It is therefore notable that parameterizations developed for global use such as those of Prata, 1996, and Dilley and O'Brien, 1998, provide the best results even for THAAO environmental conditions in both their original and site-optimized versions.

Any conclusion or direction to provide generalization of the results would be for sure very much appreciated by the community. In particular, to what extent the tuned parameters for a site (THAAO for example) can be of relevance for other sites or not can be discussed by the authors.

The following sentence has been added to the conclusions, line 431:

*The coefficients of the parametrization optimized for the 2017 and 2018 meteorological conditions at the THAAO (see paragraph 4.1 Dataset characteristics) are provided in Table 2; their performance and therefore their applicability to other sites is mostly related to the meteorological condition. The more they are similar*

*to THAAO, the more the parameterizations found for THAAO can be directly applied with similar uncertainties (e.g. RMSE value).*

SUMMPLEMENT MATERIAL

SEASONAL VARIABILITY OF CLOUD-FREE OCCURRENCE AND PERFORMANCE OF THE ANNUAL PARAMETERIZATIONS FOR THE MONTHS OF JANUARY-FEBRUARY AND JULY-AUGUST OF BOTH 2017 AND 2018.

Figure 1b shows the monthly availability of the data necessary to carry out the study calculated using the number of minutes contained in each month. Even in the months with more data, the coverage does not exceed 85% mainly due to the measurements of the microwave radiometer that has scheduled interruptions of the IWV measurements to carry out internal calibrations and measurements at different zenith angles necessary to estimate the temperature profile.
As can also be deduced from the time series shown in Figure 1 of the manuscript, the periods with the greatest interruptions concern November and December 2017, caused by the missing of data first of the IWV and then of DLI and IBT, and of September 2018 due to the lack of DLI and IBT data, which are acquired by the same datalogger. In January and December 2017 there is a reduced percentage of observations also due to the snowfall accumulation on the pyrometer or pyrgeometer or both (see paper lines 184-189), which was manually assessed by looking at the data.

The occurrence of cases of clear skies was evaluated both with respect to the total number of minutes present in each month and with respect to the performed measurements; they are called *monthly clear sky* and *normalized monthly clear sky*, respectively.
Using the *normalized monthly clear sky* (discussed in the present work), the strongest annual variations in the clear sky occurrence in the two years are found in April and June-July, months that in both 2017 and 2018 have presented a high number of observations.
The difference observed between April 2017 and 2018 constitutes an anomaly compared to the clear sky occurrence observed in March and June, which are substantially the same in the two years.
The situation is different for the differences shown between June-July 2017 and 2018. In fact, the lower value of the clear sky in June 2017 compared to that of 2018, is compensated by the higher value in July 2017 compared to that of 2018, suggesting a different temporal development of the summer season.
For the remaining months, the *normalized monthly clear sky* is very similar for the two considered years.

[Figure]

*Figure 1b. From bottom to top: time series of monthly data availability, of the monthly occurrence of clear skies calculated considering all the measurable data and of the normalized monthly occurrence of clear skies calculated with respect to the data measured.*

Overall, both the monthly availability of the data used and the actual occurrence of clear skies during the two years are homogenous in relation to the variability of the parameters in Figure 3 of the manuscript, showing that the results are affected by reduced monthly representativeness of the measures.

As suggested by the reviewer, the effectiveness of the parameterizations optimized in 2017 and 2018 was tested for different periods of the year. Based on data temporal representativeness, we have chosen to verify the performances by grouping the months of January-February and July-August representative of the winter and summer periods, respectively. To present the results of the annual parameterization applied to different periods of the year, we show the bias and the RMSE values separately, reporting in Figure 2b the performances evaluated on the whole year, on the winter and summer periods, for both 2017 and 2018.

Some particularly high RMSE values (corresponding to parameterizations of ID#3, Swinbank (1963), and ID#6, Brutsaert (1975), in 2017 and of ID#6 in 2018), or bias values very far from 0 ( ID#1, i.e. Maykut and Church (1973), ID#3 and ID#6 in both 2017 and 2018), do not appear in Figure 2b because they are very high and representative of parameterizations with low performance as already highlighted in the text of the manuscript. This choice was made to highlight the better performance of the other parameterizations.

[Figure]

*Figure 2b. From bottom to top: on the left the values of RMSE and bias for the annual (black dot), winter (blue circle) and summer (red circle) data of 2017; on the right, the same for 2018. The performances were evaluated against the optimized parameterizations on annual data from 2017 and 2018 respectively. Some values of the RMSE of the parameterizations ID#3 (Swinbank, 1973), ID#6 (Brutsaert, 1975) in 2017 and of ID#6 in 2018, as well as some values of the bias of ID#1 (Maykut and Church, 1973), ID#3 and ID#6 in both 2017 and 2018, are out of scale of the plot (see comment in the text).*

In terms of bias, considering both the seasonal differences of 2017 and 2018, no single parameterization always presents the best performance. In 2017 both ID#5, Ohmura (1981), and ID#11, Jin et al. (2006), show very small summer and winter values close to the annual ones. On the other hand, in 2018, ID#7, Satterlund (1979), shows this behavior, while ID#11 shows practically zero summer and annual bias, but a winter bias slightly larger than 2 W/m$^2$. The parameterizations of ID#12, Prata (1996), ID#13, Dilley and O'Brien (1998), and ID#14, Dilley and O'Brien (1998), have annual bias very close to 0 and present respectively positive and negative biases in the summer and winter seasons (ID#12 in 2017, and ID #16 and ID#17 in 2018), or only negative for ID#12 in 2018 or only positive for ID#16 and ID#17 in 2018.

Regarding the seasonal bias, there is no homogeneous behavior between the parameterizations.

Considering the RMSE, in 2017, there is a behavior shared by practically all stations: the winter RMSE values are lower than the annual ones, and the summer ones are approximately equal to or slightly higher than the annual ones. It is interesting to note that in 2018 the RMSE values do not present this feature, for various parameterizations (ID#2, ID#7, ID#8, i.e. Idso (1981) and Andreas and Ackley (1994), ID#11, ID#12, ID#16 and ID#17) the summer value being less than or equal to the annual one.

The parameterizations ID#11, specifically developed for the Arctic and using both the values of $e_s$ and $T_s$, ID#12, ID#16 and ID#17, generally show the lowest values of RMSE. Although the ID#12, ID#16 and ID#17 parameterizations use the values of $e_s$, $T_s$ and IWV, it is notable that having been formulated for a global application, obtain these good results when applied to the peculiar conditions of the arctic region.

However also parametrizations ID#2, ID#7, and ID#8, show very good performance using respectively only $e_s$, ID#2, or $e_s$ and $T_s$ (ID#7 and ID#8).

The best performances are maintained around RMSE values between 4 and 6 W/m$^2$ and, therefore, are statistically comparable with the accuracy of the pyrgeometer measurement.

In conclusion, we want to highlight that the RMSE is calculated as the quadratic sum of the bias and the standard deviation (see paragraph 4.2 of the manuscript), therefore parameterizations that present a non-negligible bias (e.g. ID#16 and ID #17) and low RMSE have a lower standard deviation, indicating the ability to better follow the variability of the measurements, compared to the parameterizations showing a lower bias, but higher RMSE values.

---

## Author Response (AR2)

Dear Dr. Ehrlich,

thank you for the explanation concerning the inclusion of corrections in the text, that we have carried out trying not to further burden the manuscript.

- Reviewer 1: Could you explain in more detail (more quantitatively) the accuracy of your clear-sky detection method?....
Please provide at least the conclusions and major results of your additional study and discussion in the manuscript (would fit into section 3).

The answer was summarized in the manuscript as in the following (line 181-200 of the manuscript):

*The choice to use the one-hour interval for the definition of clear sky is based on a preliminary analysis of the database, and it is in line with the approach followed by Dupont et al. (2008) who used hourly lidar averages, for comparing clear sky values derived from shortwave and longwave measurements with those derived from lidar measurements.*
*To evaluate the sensitivity of the implemented methodology to identify clear sky cases in the presence of thin cirrus clouds, both the IBT and the DLI were simulated for cloud-free conditions and with an homogeneous cirrus cloud, by means of the MODTRAN5.3 radiative transfer model (Berk et al., 2006). Different cloud optical thickness values were assumed, from 0.03 to 5, both in winter and summer conditions. The results of simulations are presented and discussed in the supplementary material. In general, the simulations highlighted the greater sensitivity of the pyrometer measurements compared to those of the pyrgeometer, particularly for low values of IWV. Based on our simulations a cirrus cloud with optical thickness of 0.1 covering homogeneously the sky in winter, induces an increase in the IBT e DLI signals compared to those for clear sky condition of 11.3 K and 2.7 W/m$^2$ respectively, corresponding to a percentage increase of 7.1% for the IBT and 1.6% for the DLI. For a similar summer case there would be an increase of 5 K and 2.7 W/m$^2$, respectively, corresponding to a percentage increase of 2.6% for the IBT and 0.97% for the DLI. These results confirm that the applied methodology is accurate enough to evaluate cases of ZCSC, even for cirrus clouds with optical thickness lower than 0.08-0.1. It should be noted that cirrus clouds of this optical thickness covering uniformly the sky induce variations in the DLI that are lower than the uncertainty of the DLI measurements, i.e. ±5 W/m$^2$, confirming that our clear sky methodology is sufficiently accurate to identify the DLI variation induced by clouds. The results of our simulations agree with those presented by Dupont et al. (2008), who highlighted that the DLI clear sky detection algorithm derived from DLI measurements perform correctly for cloud optical thickness of 0.3 or less, also evidencing that tall, thin clouds may not be detected by pyrgeometer measurements.*

Furthermore, part of the answer to reviewer 1 was also added to the other supplementary material under the title "On the sensitivity of the implemented clear sky retrieval on the presence of the thin cirrus and the relations between zenith clear sky and clear sky".

- Reviewer 2: Section 6.2: can more details be provided on the procedure to estimate the optimized coefficients? Please do so in the manuscript and not only in the replies.

The reply was summarized and insert in the paper as follow (lines 348-356 of the manuscript):

*To estimate the coefficients of the formulas using data measured at THAAO, the functions of the IDL® Software version 7.1.2 were used. Table 2 lists the functions used for each formula; details of the functions used can be found on the web page https://www.nv5geospatialsoftware.com/docs/routines-1.html. All DLI formulas are derived by emissivity parameterizations except for ID#13, ID#16, and ID#17; with the exception of these cases, the emissivity was first calculated, and the performances were then obtained by comparing the measured and parameterized values of the DLI. Taking into account the size of the database, the functions were applied without considering any uncertainty in the measurements.*

*Table 2. List of the function of IDL® Software used to derive the THAAO optimized coefficients.*

| ID# formula | IDL function |
| --- | --- |
| All ID# except those listed below | LADFIT |
| ID#11 | POLY_FIT |
| ID#16 and ID#17 | REGRESS |
| ID#7 and ID#12 | CURVEFIT |

- Reviewer 2: Section 7, general comments: based on the analysis and conclusions of the paper, two points can be raised and maybe deserve a word of conclusion from the authors...
Again, include your discussion of the replies in the revised manuscript.

We have discussed the points highlighted by the reviewer 2 introducing at the beginning of the conclusion the following short discussion concerning the estimation of DLI and its dependence from the measured parameters (lines 437-448 of the manuscript):

*In clear sky conditions the DLI is determined by the atmospheric concentration of greenhouse gases (mainly water vapor) in the lowest atmospheric layers and by their radiant temperature. The most reliable estimate of DLI is obtained by accurate measurements of the water vapor and temperature profiles, e.g. from radiosounding, used as input for radiative transfer models: however, this methodology is unpractical for large datasets, thus the need for DLI parameterizations.*

*The existing formulas need to be tuned to the environmental conditions of the application region because DLI estimate not from atmospheric profiles, but from proxies, such as screen-level measurements (e.g. air temperature or/and water vapour partial pressure) and/or integrated water vapor, were derived for a range of site-specific meteorological conditions. The effectiveness of each formula therefore depends on the representativeness of the parameters it uses to represent the atmospheric vertical profile and on its ability to simulate the emission processes in the lower layers of the atmosphere, which determine the DLI value. From this perspective formulas derived from the use of radiative transfer models are therefore expected to have more general validity than purely empirical ones.*

In the conclusion we have also pointed out the role of the parameters used in the formulas and the applicability of the obtained results in other sites/conditions, see author's track-changes file (in particular see lines 492-496 of the manuscript)

*The analyses carried out on two different meteorological years indicate that the optimization of a formula carried out on one year is applicable to different years producing an increase in RMSE of ~1-2 $W/m^2$ depending on the selected formula; these results are strictly valid for the database used. This analysis suggests that for other sites the formula optimised on a reference year can be applied to other years with similar variability of atmospheric conditions, with an expected small increase in RMSE.*